# An In-Depth Study of Metabolite Profile and Biological Potential of *Tanacetum balsamita* L. (Costmary)

**DOI:** 10.3390/plants12010022

**Published:** 2022-12-20

**Authors:** Reneta Gevrenova, Gokhan Zengin, Kouadio Ibrahime Sinan, Dimitrina Zheleva-Dimitrova, Vessela Balabanova, Maxime Kolmayer, Yulian Voynikov, Olivier Joubert

**Affiliations:** 1Department of Pharmacognosy, Faculty of Pharmacy, Medical University-Sofia, 1431 Sofia, Bulgaria; 2Department of Biology, Faculty of Science, Selcuk University, Konya 42250, Turkey; 3Institut Jean Lamour, Unité Mixte de Recherche Centre National de la Recherche Scientifique 7198, Université de Lorraine, F-54000 Nancy, France; 4Department of Chemistry, Faculty of Pharmacy, Medical University-Sofia, 1431 Sofia, Bulgaria

**Keywords:** *Tanacetum balsamita*, secondary metabolites, antioxidant properties, enzyme inhibitory activity, cytotoxic activity, UHPLC-HRMS

## Abstract

Asteraceae species *Tanacetum balsamita* L. (costmary) is renowned for its traditional usage as an aromatic, carminative and tonic plant. This work aimed at in-depth study of the phytochemical and in vitro biological profilings of methanol–aqueous extracts from the costmary leaves, flower heads and roots. An UHPLC-HRMS analysis revealed more than 100 secondary metabolites including 24 acylquinic acids, 43 flavonoid glycosides, aglycones and methoxylated derivatives together with 15 phenolic acids glycosides. For the first time, 91 compounds are reported in the costmary. The flower heads extract possessing the highest content of total phenolics and flavonoids, actively scavenged DPPH (84.54 ± 3.35 mgTE/g) and ABTS radicals (96.35 ± 2.22 mgTE/g), and showed the highest reducing potential (151.20 and 93.22 mg TE/g for CUPRAC and FRAP, respectively). The leaves extract exhibited the highest inhibition towards acetyl- and butyrylcholinesterase (2.11 and 2.43 mg GALAE/g, respectively) and tyrosinase (54.65 mg KAE/g). The root extract inhibited α-glucosidase (0.71 ± 0.07 mmol ACAE/g), α-amylase (0.43 ± 0.02 mmol ACAE/g) and lipase (8.15 ± 1.00 mg OE/g). At a concentration >2 µg/mL, a significant dose dependent reduction of cell viability towards THP-1 monocyte leukemic cells was observed. Costmary could be recommended for raw material production with antioxidant and enzyme inhibitory properties.

## 1. Introduction

The use of plants as sources of drugs and secondary metabolites has been attracting scientific attention over the past decades, considering not only the well-known medicinal species but also plants used in traditional medicines and a variety of edible plants. *Tanacetum balsamita* L. (costmary) is renowned for its traditional use as flavor, carminative and cardiotonic in the Mediterranean, Balkan and South American countries [1]. The species is distributed in the South-East of Europe and South-West of Asia but has also been widely naturalized throughout the whole world [2,3]. *T. balsamita* is commonly referred to as: costmary, balsam herb, alecost, sweet tongue and bible leaf. The plant is cultivated in Iran, Turkey, Romania, Germany, Italy, Spain and England [1]. It has a traditional usage as aromatic plant in Europe and Asia. Fresh and dried costmary leaves possess a strong lemony-minty flavor and a sweet astringent taste. The dried leaves have a long history of application as flavorings in soups and meats, sausages and cakes, as well as for making tonic tea. Costmary leaves have been used in ethnopharmacological approach as a hepatoprotective, tonic, sedative, pain relief and astringent agent [1]. The analgesic, anti-inflammatory, antimicrobial and antioxidant activity of the essential oil and extracts support the traditional claims regarding the costmary use for the alleviation of inflammatory diseases [4,5,6,7,8]. Venskutonis [7] and Hassanpouraghdam et al. [1] reviewed the phytochemistry and application of costmary and highlighted that it may be considered as a forgotten plant with potential for development of functional food ingredients.

Costmary is well known for the essential oil secreted in the glandular trichomes. Overall, a total of 186 compounds have been reported in the essential oils from aerial parts, flower heads and stems [7]. According to their main constituents, four chemotypes were distinguished: carvone, camphor, camphor–thujone, and camphor-α-thujone [1]. Moreover, bornyl acetate-pinocarvone chemotype of *T. balsamita* spp. balsamitoides was established by Jaimand and Rezaee [9]. A high level of carvone was evidenced in costmary leaves essential oils (51–80% in the total oil content) [6,8,10,11,12,13], while α-thujone and β-thujone reached 16% and 84%, respectively [11,14]. Even though the various protective effects of carvone, being the most abundant constituents in the majority of the costmary essential oils, it is worth noting that α-thujone and β-thujone are toxic monoterpenes. Thus, great care should be taken regarding the internal use of α-thujone and β-thujone rich essential oils of costmary. EMA (European Medicinal Agency) and EC (European Commission) recommended a maximum uptake between 3 and 7 mg thujone/day [15]. The cytotoxicity of *T. balsamita* spp. *balsamita* essential oil towards human fetal skin fibroblast (HFSF) and monkey kidney cells (Vero) expressed as IC50 was 2500 ± 1.9 µg/mL and 1250 ± 1.4 μg/mL, respectively [8].

Costmary contains caffeoylquinic acids, methoxylated flavonoids and sesquiterpene lactones [5,14,16,17]. For instance, chlorogenic acid, 3, 5-dicaffeoylquinic acid, 5, 7, 4’-trihydroxy-3’,8-dimethoxyflavon and 5,7,3’,4’-tetrahydroxy-3,8-dimethoxyflavonol were isolated from costmary aerial parts [17]. Benedec et al. [16] and Baczek et al. [14] determined by HPLC-MS/DAD several caffeic acid derivatives and flavonoids in the ethanol-aqueous extracts from aerial parts. Cichoric acid dominated the polyphenols, being present in 3.33 g/100 g dry extract [14], while rutin reached 8.04 mg/g dry material [16]. Diosmetin and acacetin glycosides were also reported [5]. Sesquiterpene lactones of eudesmanolide and C-9β-hydroxylated or esterified germacranolide type have been isolated from cultivated costmary [18,19].

In vitro antioxidant, antimicrobial and cytotoxic activity of costmary essential oils and extracts has been reported [5,7,8,14,16]. Overall, the authors concluded that costmary had a great potential for pharmaceutical and food industry, notably for nutraceuticals.

It is worth noting that the aforementioned studies highlight the essential oil composition and antimicrobial effect, while there is no in-depth metabolic profile of costmary leaves and flower heads by liquid chromatography-high-resolution mass spectrometry (LC-HRMS) integrated with an assessment of antioxidant, enzyme inhibitory and cytotoxic potential. Recently, metabolic and biological profilings of *T. macrophyllum* and *T. vulgare* integrated with multivariate data analysis provided a new insight into the taxa for designing health-promoting applications [20,21].

Species of the *Tanacetum* genus are of economic importance. *T. parthenium*, *T. macrophyllum* and *T. vulgare* are cultivated worldwide as ornamental plants and raw material for food, pharmaceutical and agricultural use [22,23,24,25].

As a part of our ongoing investigation on *Tanacetum* [20,21,25], in this study we aimed at an in-depth metabolite and biological profiling, including antioxidant, enzyme inhibitory and cytotoxic activity of methanol–aqueous extracts from *T. balsamita* leaves, flower heads and roots. Multivariate data analyses were applied to determine possible correlations between chemical compositions and results from the biological assays of the tested extracts.

## 2. Results and Discussion

### 2.1. Secondary Metabolite Profiling of Tanacetum Balsamita Extracts

Based on our previous study on the solvent efficiency in the phenolic and flavonoid extraction from *T. vulgare* [20], methanol–aqueous extracts from *T. balsamita* flower heads, leaves and roots were obtained and the total phenolic (TPC) and flavonoid (TFC) contents were determined (Table 1). The highest TPC and TFC were found in the flower heads (59.75 ± 0.66 mg GAE/g and 41.02 ± 0.50 mg QE, respectively), while the lowest values were determined in the leaves (30.82 ± 0.16 mg GAE/g for TPC) and roots (3.74 ± 0.07 mg QE for TFC). It is worth noting that the results revealed a higher amount in both studied classes compared to those in *Chrysanthemum balsamita* var. *tanacetoides* aerial parts, where 2.92 g GAE% and 1.19 g RE% were evidenced [16]. Overall, our results were in the same order of magnitude or lower than TPC and TFC reported in flower heads and aerial parts from *T. poterifolim*, *T. macrophyllum*, *T. vulgare* and *T. parthenium* with a Turkish and a Bulgarian provenance [20,21,25,26] and higher than those in *T. corymbosum*, *T. vulgare* and *T. macrophyllum* from Romanian origin [27]. In line with the results, the accumulation of phenolic compounds could be firmly connected to the plant ontogenetic development and they could be related differently to each plant organs [28]. Therefore, the diverse qualitative and quantitative content of the bioactive compounds reveal different potential and effects.

As it was previously reported, *Tanacetum* species contain a wide range of hydroxycinnamic esters and flavonoid derivatives [20,21,24,25,26,29]. To assess the secondary metabolites, non-targeted metabolic profiling of the hydroxycinnamic acids, flavones, flavonols and flavanones of each methanol–aqueous extract was carried out by UHPLC-Orbitrap-HRMS. The parameters of Full-scan ddMS^2^ mode were adjusted to favor the formation of diagnostic fragment ions for the subclasses of phenolic acids derivatives, acylquinic acids (AQA) and flavonoids. Fragmentation patterns along with exact masses of precursor ions in negative and positive (for flavonoids) ionization mode were depicted in Table 2 and Appendix A. Thus, based on the fragmentation patterns and characteristic ions and authentic standards, fragmentation keys for recognition of AQA and methoxylated flavonoids were generated.

#### 2.1.1. Hydroxybenzoic and Hydroxycinnamic Acids and Their Derivatives

Based on the comparison with the fragmentation patterns and retention times of reference standards, 5 hydroxybenzoic acids (**3**, **6**, **13**, **16** and **33**) and 4 hydroxycinnamic acids (**27**, **32**, **40** and **41**) together with *p*-hydroxyphenylacetic (**31**) and rosmarinic acid (**52**) were identified in the extracts (Table 2 and Appendix A).

A variety of hydroxybenzoic and hydroxycinnamic acid glycosides was tentatively identified including hexosides (**1**, **4**, **5**, **7**, **9**, **11**, **17**, **18**, **21**, **24**, **25**, **26** and **39**) along with pentosylhexoside of hydroxybenzoic acid (**2**) and dihydroxyphenylacetic acid (**19**). MS/MS spectra of sugar esters vanillyl-hexose (**10**) and caffeoyl-hexose (**15**) were acquired. In contrast to the corresponding hexosides, fragment ions resulting from the sugar cross ring cleavages were registered in the sugar esters as follows: ^0,4^Hex (−60 Da), ^0,3^Hex (−90 Da) and ^0,2^Hex (−120 Da) [20]. In addition, three caffeoylgluconic acid isomers (**14**, **21** and **29**) at *m*/*z* 357.084 [M-H]^−^ were deduced from the prominent ions at *m*/*z* 195.050 [gluconic acid (GA)-H]^−^ supported by *m*/*z* 177.040 [GA-H-H_2_O]^−^, 165.040 [GA-H-CH_2_O]^−^, 147.028 [GA-H-CH_2_O-H_2_O]^−^, 129.018 [GA-H-CH_2_O-2H_2_O]^−^, 87.007 [GA-H-C_3_H_8_O_4_]^−^ and 59.012 [GA-H-C_3_H_8_O_4_-CO]^−^ (Table 2). Within this group, **25** was the main compound in the leaves extract, together with **6**, **1**, **22** and **46**, while roots and flower heads were dominated by **22** (Appendix A). Even though hydroxybenzoic and hydroxycinnamic acids were present in their free form, herein a large number of phenolic acid glycosides were revealed in costmary for the first time.

Previously, cichoric acid was determined as prevailing compound in the costmary aerial parts, being present at 3.33 g/100 g extract [14]. In contrast, cichoric acid was not found in this study. Ethanol-aqueous extracts of *T. vulgare* leaves and flower heads were especially rich in protocatechuic acid and its hexoside along with caffeic and salicylic acid and caffeic acid-*O*-(salicyl)-hexoside [20]. In line with these findings, *T. parthenium* aerial parts were rich in *p*-hydroxyphenylacetic and caffeic acid, being present in 280.4 and 129.8 mg/kg extract, respectively [25]. On the other hand, *T. macrophyllum* was distinguished by the phenylpropanoid glycosides caffeic acid-*O*-(hydroxybutanoyl)-hexoside and vanillic/gentisic acid-*O*-(caffeoyl)-hexoside, together with two caffeoyl-(syringic) acid isomers.

#### 2.1.2. Acylquinic Acids

Overall, 8 *mono*AQA, 13 *di*AQA and 1 *tri*AQA, together with two *di*AQA-hexosides, were identified/annotated in the assayed extracts (Table 2 and Appendix A). The systematic investigation on the fragmentation patterns and diagnostic fingerprints in the MS/MS spectra of AQA in Asteraceae taxa allowed for the differentiation of the AQA subclasses [20,21,30]. The AQA annotation was based on the preferential fragmentation resulting in relevant ions corresponding to each subclass AQA. Thus, **23**, **34/42** and **36** were ascribed as 5-caffeoyl-, 5-coumaroyl- and 5-feruloylquinic acid, respectively, by the base peak at *m*/*z* 191.055 [quinic acid-H]^−^. Compounds **8** and **30** were identified as 3-caffeoyl- and 3- feruloylquinic acid, while **28** and **43** were assigned to the respective 4-substituted *mono*AQA as suggested the base peak corresponding to the dehydrated ion of quinic acid at 173.045 (Table 2).

Five commonly found *di*AQA subclasses were identified/annotated: di-caffeoylquinic acid isomers (*di*CQA) at *m*/*z* 515.120 [M-H]^−^, feruloyl-caffeolylquinic acids (FCQA) at *m*/*z* 529.135, *p*-coumaroyl-caffeoylquinic acids (*p*-CoCQA) at *m*/*z* 499.125, hydroxydihydrocaffeoyl-caffeolylquinic acids (HC-CQA) at *m/z* 533.130 and dehydrocaffeoyl-caffeoylquinic acid (DC-CQA) at *m*/*z* 513.104 (Table 2).

Compounds **46**, **50**, **53**, **57**, **58** and **59** were consistent with vicinal *di*AQA yielding prominent ions at *m*/*z* 173.044 (base peak) and 135.044 [caffeic acid-H-CO_2_]^−^. The dehydrated ion at *m*/*z* 335.077 [CQA-H-H_2_O]^−^ clearly defined 3,4-*di*CQA (**46**) and 3F-4CQA (**53**). The assignment of FCQA was confirmed by the prominent fragment ions at *m*/*z* 367.103 [M-H-caffeoyl]^−^ and 134.036 [ferulic acid-H-CH_3_-CO_2_]^−^. The lack of ion at *m*/*z* 335 (or its negligible abundance) together with the chromatographic behavior on the reverse phase support (the most lipophilic *di*AQA isomer within the subclass) evidenced 4,5-*di*CQA (**50**), 4F-5CQA (**57**), 4C-5FQA (**58**) and 4C-5-*p*-CoQA (**59**). This assignment was supported by the abundant ions at *m*/*z* 367.103 [M-H-caffeoyl]^−^ (100%) (**57**), 353.088 [M-H-feruloyl]^−^ (49.7%) (**58**) and 353.087 [M-H-*p*-coumaroyl]^−^ (76.7%) (**59**) (Table 2). Generally, 3,5-*di*AQAs easily cleaves the acyl moiety at C-5, compared to C-3. Thus, **48** (3,5- *di*CQA) produced a base peak at *m*/*z* 191.055 accompanied with abundant ions at *m*/*z* 179.034 [caffeic acid-H]^−^ and 135.044, while the relevant ions at *m*/*z* 367.103 [M-H-caffeoyl]^−^ (97.1%) and 193.050 [ferulic acid-H]^−^ (base peak) suggested 3F-5CQA (**56**) (Table 2). In the same manner, **54** was ascribed as 3-*p*-Co-5CQA by the base peak at *m*/*z* 163.039 [*p*-coumaric acid-H]^−^ and diagnostic ions at *m*/*z* 337.093 [M-H-caffeoyl]^−^ (75.6%) and 119.049 [*p*-coumaric acid-H-CO_2_]^−^ (37.5%) (Table 2). 3-DC-5-CQA (**47**) was deduced from the distinctive fragments at *m*/*z* 351.074 [M-H-caffeoyl]^−^ (100%), 177.018 [dehydrocaffeic acid-H]^−^ (53.8%) and 133.028 [dehydrocaffeic acid-H-CO_2_]^−^ (86.1%), while 3-HC-5-CQA (**35**) was evidenced by ions at *m*/*z* 371.099 [M-H-caffeoyl]^−^, 191.055 and 135.044.

Peaks **44** and **45** gave a precursor ion at *m*/*z* 677.173(4) (C_31_H_33_O_17_), together with the transitions at *m*/*z* 677.173→515.141→353.088→191.055 (**44**) resulting from the losses of two caffeoyl moieties and a hexose unit, respectively (Table 2). 3,5-*di*substituted quinic acid skeleton (**44)** was discernible by the ions at *m*/*z* 191.055 (99.7%), 179. 034 (44.8%) and 135.044 (42.5%). Thus, **44** was ascribed as 3,5-*di*caffeoylquinic acid-hexoside. In the same way, 4,5-*di*caffeoylquinic acid-hexoside (**45**) was deduced from the ions at *m*/*z* 173.044 (71.6%) and 179.034 (60.5%), together with *m*/*z* 341.089 and 323.079.

Compound **60** with [M-H]^−^ at *m*/*z* 677.152 (consistent with C_34_H_29_O_15_) was the most hydrophobic AQA. It afforded prominent fragment ions at *m*/*z* 515.120 [M-H-caffeoyl]^−^, 353.089 [M-H-2caffeoyl]^−^ and 191.055 [M-H-3caffeoyl]^−^ indicating *tri*CQA. 3,4,5- *tri*CQA was discernible from the fragment ions at *m*/*z* 173.044 (100%), 135.044 (82%) and 179.034 (76.8%) [31].

Overall, **23**, **28** and **36** were the major *mono*AQA together with *di*AQA **46**, **48** and **50** (Appendix A). These results were consistent with the AQAs evidenced in *T. vulgare* and *T. parthenium* aerial parts extracts where chlorogenic acid was found between 2.6 mg/g dw and 925 mg/100 g extract, and 487.8 mg/kg extract, respectively [14,25,27]. *T. macrophyllum* was discernible by 1,5 diCQA, isomeric pairs 4-*p*-Co-5-CQA/4C-5-*p*-CoQA and 1C-5FQA/1C-3FQA, together with a variety of HC-CQA and DC-CQA [20,21].

#### 2.1.3. Flavones, Flavonols and Flavanones

MS/MS spectra of **61** and **62** at *m*/*z* 595.168 [M-H]^−^ and 593.152, respectively, were acquired (Table 2 and Appendix A). A typical di*C*-hexosyl flavonoid fragmentation pathway was observed including a series of the following transitions: [M-H-120]^−^ at *m*/*z* 475.126 (**61**) and 473.109 (**62**), [M-H-120-60]^−^ at *m*/*z* 415.104 and 413.089, [M-H-120-90]^−^ at *m*/*z* 385.093 and 383.078, and [M-H-2 × 120]^−^ at *m*/*z* 355.083 and 353.067, respectively (Table 2). Additionally, the aglycone naringenin in **61** was evidenced by the deprotonated molecule at *m*/*z* 271.062 supported by the RDA ions at *m*/*z* 163.003 (^0,2^A^−^), 151.002 (^1,3^A^−^), 119.049 (^1,3^B^−^) and 107.012. (^0,4^A^−^). Concerning **62**, the aglycone apigenin was discernible by the prominent ions at *m*/*z* 325.071 [(M-H)-2 × 120-CO]^−^, 297.077 [(M-H)-2 × 120-2CO]^−^ and 117.033 (^1,3^B^−^) in (-) ESI-MS/MS supported by *m*/*z* 177.018 [^1,3^A^+^-H_2_O (^0,2^X_0_/^0,2^X_1_)] and 121.028 [^0,2^B^+^ (^0,2^X_0_/^0,2^X_1_)] (Table 2 and Appendix A). Thus, **61** and **62** were ascribed as 6,8-di*C*-hexosyl-naringenin and 6,8-di*C*-hexosyl-apigenin (vicenin-2), respectively. In the same way, 6-*C*-hexosyl-luteolin was confirmed by the diagnostic ions ^0,3^X^−^ at *m*/*z* 357.061 (39.3%) and ^0,2^X^−^ at *m*/*z* 327.051 (53.7%); luteolin ([Lu-H]^−^ at *m*/*z* 285.041) was deduced from RDA ions ^1,3^B^−^ at *m*/*z* 133.028 and ^1,4^A^−^ at *m*/*z* 175.038. Based on the comparison with reference standards, **62** and **63** were unambiguously identified as vicenin-2 [32] and homoorientin, respectively.

The sugar chain of **66**, **69**, **76**, **77**, **80** and **81** was consistent with pentosylhexoside (294 Da, C_16_H_18_O_9_) (Table 2 and Appendix A). Exemplified by **76**, in (-) ESI, the precursor ion at *m*/*z* 639.157 [M-H]^−^ gave the prominent ion at *m*/*z* 345.062 corresponding to the deprotonated aglycone, while in (+) ESI [M+H]^+^ at *m*/*z* 641.167 exhibited successive losses of pentose moiety at *m*/*z* 509.127 (31.5%) and hexosyl at *m*/*z* 347.076 (100%). On the other hand, the aglycone (Agl) was discernible by the consecutive methyl radical losses at *m*/*z* 330.039 [Agl-H-•CH_3_]^−^, 315.015 [M-H-2•CH_3_]^−^, 287.019 [M-H-2•CH_3_-CO]^−^ in (-) ESI supported by the transitions 347.076→331.045→289.046 in (+) ESI. Thus, **76** was referred to quercetagetin-dimethyl ether-*O*-pentosylhexoside.

Compounds **65**, **70**, **74** and **78** were closely associated to the same fragmentation pattern giving characteristic fragment ions at *m*/*z* 301.035 (**65**), 285.040 (**70**, **74**) and 269.045 (**78**) [(M-H)-308]^−^, indicating deoxyhexosylhexosides (Table 2). In (+) ESI, an interglycosidic linkage breakdown occurred, yielding [M-H-146] at *m*/*z* 465.102 (**65**), 449.108 (**70**, **74**) and 433.112 (**78**) (Appendix A), as was observed in case of 1→6 linkage [33,34]. Thus, aforementioned compounds were assigned as quercetin/luteolin/kaempferol/apigenin-*O*-rutinosides.

Compounds **72**, **73**, **83**, **85**, and **86** presented similar fragmentation patterns yielding base peaks at *m*/*z* 285.040 (**72**), 315.051 (**73**), 269.045 (**83**), 329.067 (**85**) and 299.056 (**86**) [(M-H)-HexA]^−^, respectively, indicating flavonoid hexuronides (Table 2 and Appendix A). Compounds **72**, **73** and **83** were consistent with luteolin-, isorhamnetin- and apigenin-*O*-hexuronide, respectively. Compound **86** was ascribed to chrysoeriol-*O*-hexuronide (^1,3^A^−^ at *m*/*z* 151.002, ^0,4^A^−^ at *m*/*z* 107.013), while **85** was assigned as jaceosidin-*O*-hexuronide [35].

Isoquercitrin (**67**), hyperoside (**68**), luteolin 7-glucoside **(71**), isorhamnetin 3-glucoside (**79**) and kaempferol 3-glucoside (**84**) were unambiguously identified by comparison with reference standards.

Hexuronides of luteolin, isorhamnetin, apigenin, chrysoeriol and jaceosidin were exclusively produced by *T. balsamita* leaves (Appendix A), Flavonoid glycosides profile of both leaves and flower heads extracts were dominated by rutin; the former was also characterized by luteolin/jaceosidin-hexuronide, while the latter was rich in isoquercitrin and hyperoside. Previously, apigenin- and luteolin 7-glucoside were determined in *T. balsamita* aerial parts, being present in 1099.3 and 725.7 mg/100 g extract [14].

### 2.2. Methoxylated Flavonoids

Methoxyflavonoids annotation was based on the characteristic fragment ions delineated in previous studies on *Tanacetum* sp. [21,30] and *Achillea* sp. [30]. Generally, the initial RDA ions ^1,3^A^−^ and ^1,3^B^−^ were not observed [35,36]. In (-) ESI, a series of RDA ions originating from ^1,3^A^−^ were produced including [^1,3^A-•CH_3_]^−^, [^1,3^A-H_2_O-CH_2_]^−^, [^1,3^A-CO-CH_2_]^−^ and [^1,3^A-CO-CH_4_]^−^, while [^1,3^B-•CH_3_-CH_2_]^−^, [^1,3^B-CH_4_]^−^ and [^1,3^B-CH_2_]^−^ suggested methoxylation in a B ring (Table 2). In (+) ESI, a commonly found fragment at *m*/*z* 168.005 [^1,3^A-•CH_3_]^+^ suggested methoxylation in an A ring (Appendix A).

MS/MS spectra of five quercetagetin derivatives (**91**, **93**, **94**, **97** and **101**) were acquired (Appendix A). The methoxylated derivative **91** was deduced from the fragment ions at *m*/*z* 316.022 [M-H-•CH_3_]^−^ and 287.020 [M-H-•CH_3_-CHO•]^−^; 6-methoxylation in a ring A was revealed by the prominent ions in (-) ESI at *m*/*z* 181.0134 [^1,3^A]^−^, 165.990 [^1,3^A-•CH_3_]^−^, 139.002 [^1,3^A-CO-CH_2_]^−^, 136.986 [^1,3^A-CO-CH_4_]^−^ and 109.999 [^1,3^A-•CH_3_-2CO]^−^, supported by *m*/*z* 168.005 [^1,3^A-•CH_3_]^+^ in (+) ESI (Table 2 and Appendix A). Compounds **93** and **94** shared the same [M-H]^−^ at *m*/*z* 345.061, indicating an additional methyl group compared with **91**. Regarding **94**, RDA ions generated from [^1,3^A]^−^ were consistent with those in **91** (Table 2). Fragment ions at *m*/*z* 121.028 [^1,2^B]^−^ and 137.023 [^0,2^B]^+^, evidenced the lack of methoxy group in a ring B. In line with this assumption, methoxy group at C-3 was suggested; accordingly, **94** was assigned as quercetagetin-3, 6-dimethyl ether (axillarin), previously reported in *Tanacetum* sp. [20,21]. Concerning **93**, a methoxylated B ring was discernible by *m*/*z* 161.023 [^1,3^B]^−^ as was observed in spinacetin.

Both isomers **97** and **101** at *m*/*z* 359.078 [M-H]^−^ generated prominent ions at *m*/*z* 344.054, 329.031 and 314.007, resulting from the consecutive losses of three methyl radicals •CH_3_. They were referred to quercetagetin 3, 6, 3’(4’)- trimethyl ether; two methoxy groups in RDA ion [^1,3^B]^−^ were evidenced by the typical losses at *m/z* 148.015 [^1,3^B-•CH_3_-CH_2_]^−^, 161.022 [^1,3^B-CH_4_]^−^ and 163.039 (^1,3^B^−^-CH_2_) (Table 2).

In the same way, four closely associated 6-methoxyluteolin derivatives (**92**, **99**, **100** and **103**) were described (Appendix A). Among them, **99** and **100** shared the same [M-H]^−^ at *m*/*z* 329.067. Jaceosidin (6-hydroxyluteolin-6, 3’-dimethylether) (**99**) was discernible by the RDA ions in (-) ESI at *m*/*z* 163.002 (^1,3^A^−^-H_2_O), 136.988 [^1,3^A-CH_4_-CO]^−^, 135.008 (^1,3^A^−^-H_2_O-CO-CH_2_) and 133.028 [^1,3^B-CH_2_]^−^, supported by *m*/*z* 168.005 [^1,3^A-CH_3_]^+^ in (+) ESI (Table 2 and Appendix A). Additionally, **99** was confirmed by comparison with reference standard. On the other hand, **100** afforded RDA fragments in (-) ESI at *m*/*z* 161.023 [^1,3^A^−^-CH_4_-H_2_O]^−^ and 151.003 [^1,3^A^−^-CH_4_-CO]^−.^together with *m*/*z* 136.015 [^1,3^A-CH_3_-H_2_O-CO]^+.^and 137.022 [^0,2^B]^+^ in (+) ESI (Table 2 and Appendix A). Accordingly, **100** could be associated with cirsiliol (6-hydroxyluteolin-6, 7-dimethylether). Compound **103** afforded prominent fragment ions at *m/z* 328.059, 313.036 and 298.012, resulting from the consequent losses of 3 methyl radicals (Table 2). A methoxylated ring A was evidenced by RDA ions at *m*/*z* 136.987 (^1,3^A^−^-CO-CH_4_) and 163.002 (^1,3^A^−^-H_2_O), while *m*/*z* 132.0203 (^1,3^B^−^-•CH_3_-CH_2_) indicated 2 methoxy groups either in C-3, C-4′ or C-3′, C-4′, as was observed in santin and eupatilin, respectively [20,37].

Two scutellarein (6-hydroxyapigenin) derivatives **94** and **96** were discernible from the prominent ions in (-) ESI at *m*/*z* 165.990 [^1,3^A-CH_3_]^−^ and 136.987 [^1,3^A-CH_3_-CO]^−^ (**96**), and *m*/*z* 151.002 [^1,3^A-CO-CH_4_]^−^ and 107.012 [^0,4^A-CO-CH_4_]^−^ (**94**) (Table 2). Thus, **94** was assigned as cirsimaritin, while **96** was referred to hispidulin, additionally confirmed by comparison with reference standard. The aforementioned assumption was supported by fragments at *m*/*z* 168.005 [^1,3^A-CH_3_]^+^ (**96**) and 153.018 [^1,3^A-CH_4_-CO]^+^ (**94**) in (+) ESI (Appendix A).

Methoxylated flavones and flavonols are widespread in Asteraceae species. Generally, nepetin (6-methoxy luteolin) and jaceosidin (6-hydroxyluteolin 3’,6-dimethyl ether) appeared to be characteristic for costmary leaves and root extracts; in contrast, flower heads were the richest in quercetin and isorhamnetin (Appendix A).

### 2.3. Heatmap Analysis

To gain an intuitive viewing of the metabolite contrast among the different extracts of *Tanacetum balsamita*, a heatmap was generated. Figure 1 displays the outcomes and highlights the arrangements of the groups of metabolites characterizing each extract. The red and blue color in the plot specify higher and lower metabolite amounts than the mean, respectively. As observed, the metabolites within group C were abundant in the leaves extract. In contrast, the lowest concentration of the metabolites within group A1 was exhibited by the leaves extract. Similarly, higher concentrations of the group A2 metabolites were found in the flower heads extract, while the group E metabolites were found at lower levels. On the other hand, the metabolites of the root were of low concentrations and were consolidated in the group B. A few studies have reported that the concentration of metabolites is varying in the different parts of the same species. High concentration of phenolics in leaves versus that of roots may be resulted to the presence or absence of light that impacts the phenolic contents of organs [38]. Furthermore, variation in the amount of various phenolic molecules in plants during its phenological cycle is reported by Çirak et al. [39].

### 2.4. Antioxidant Properties

Antioxidants impair the oxidative damage in foods and herbs by delaying or inhibiting oxidation, and expand the shelf-life and quality of these foods [40]. Thus, their consumption could be of help in the treatment of diseases correlated with oxidative damage, as cardiac vascular diseases, inflammations, diabetes and cancer [41]. Therefore, in the present work, the in vitro antioxidant properties of *T. balsamita* extracts were assayed, and the results are depicted in Table 3.

The collected data are consistent with the highest values of TPC and TFC (Table 1). The total antioxidant capacity (TAC) of the *T. balsamita* extracts was evaluated by the phosphomolybdenum assay, where the highest values, up to 1.48 ± 0.01 mmol TE/g were detected in the root extract, with the highest values, followed by flower heads and leaf extracts (Table 3). Regarding TAC, our results were comparable to those obtained in *T. poterifolim*, *T. macrophyllum*, *T. vulgare* and *T. parthenium* [20,21,25,26]. Additional antioxidant assays were carried out to provide insights into the antioxidant properties of the assayed *T. balsamita* extracts. The flower heads extract had the most pronounced radical scavenging activity in DPPH (84.54 ± 3.35 mg TE/g) and ABTS (96.35 ± 2.22 mg TE/g) assays. Reducing power is an important way to evaluate electron-donating ability of antioxidants. Thus, the reducing power of the assayed extracts was investigated by FRAP (from Fe^3+^ to Fe^2+^) and CUPRAC (from Cu^2+^ to Cu^+^) assays. The highest reducing power in both assays was found in the flower heads extract (93.22 ± 1.59 mg TE/g for FRAP and 151.20 ± 0.22 mg TE/g for CUPRAC). The results compare favorably with our previous study on *Tanacetum* species using the same assays. Generally, the received data for radical scavenging activity are consistent with those previously recorded in *T. parthenium, T. poteriifolium* and *T. vulgare* extracts and substantially lower in comparison with *T. macrophyllum* extracts [20,21,25,26]. The aforementioned *Tanacetum* species revealed higher reducing power activity than costmary extracts. *T. balsamita* leaves extract possessed strong chelating ability being more potent compared to *T. parthenium, T. poteriifolium* and *T. vulgare* extracts.

### 2.5. Enzyme Inhibitory Activity

The inhibitory ability of extracts prepared from the *T. balsamita* flower heads, leaves and roots against enzymes targeted in the management of type II diabetes mellitus, Alzheimer’s disease, lipid metabolism and skin hyperpigmentation problems were investigated.

The highest AChE and BChE inhibitory potential were observed for the leaf extract (2.11 ± 0.04 mg GALAE/g and 2.43 ± 0.04 mg GALAE/g, respectively) (Table 4).

The same sample was found to have the highest tyrosinase and α-amylase inhibitory activity (54.65 ± 1.30 mg KAE/g and 0.44 ± 0.01 mmol ACAE/g, respectively). α-Glucosidase inhibitory potential did not exceed 0.71 ± 0.07 mmolACAE/g, while lipase inhibition was up to 8.15 ± 1.00 mg OE/g, whre both were for the root extract (Table 4). Aforementioned results were consistent with those previously recorded in *T. vulgare* except for the lower α-glucosidase and the higher tyrosinase inhibitory potential [20]. It’s worth noting that *T. poteriifolium* aerial parts demonstrated remarkable tyrosinase and α-glucosidase inhibition among the assayed *Tanacetum* species [26].

These findings could be related to the extracts chemical profiling (Table 2). For instance, flavonoids and acylquinic acids have been shown as inhibitors of the studied enzymes. However, the enzyme inhibitory potential is not directly related to TPC and TFC as seen in *T. vulgare* and *T. macrophyllum* [20,21]. It appears that the enzyme inhibition could be ascribed to the sesquiterpene lactones. Hence, it may be assumed that sesquiterpene lactones act in a synergistic way in AChE related disorders [42]. Furthermore, the germacranolide parthenolid and monoterpene thujone have been already reported as cholinesterase inhibitors [43,44,45]. Orhan et al. (2015) hypothesized that parthenolode plays a role in AChE inhibition in a synergistic manner together with other compounds (monoterpenes). Thus, the leaves extract of *Tanacetum argenteum* subsp. *flabellifolium* had the highest AChE inhibitory effect (96.68 ± 0.35%). *C*-flavonoid glycoside homoorientin, identified in costmary leaves extract, was previously reported to inhibit AChE in an in silico and in vivo study [46]. At 100 mg/kg for 3 weeks homoorientin inhibited the activity of AChE in rats with experimentally induced Alzheimer’s disease. Hispidulin, identified in the *Phyla nodiflora* extracts, was previously reported to inhibit tyrosinase with an IC_50_ value of 146 µM [47]. In addition, chlorogenic acid and its derivatives have potential as cholinesterase and glucosidase inhibitors [48,49,50]. Thus, chlorogenic acid inhibited AChE and BChE and pro-oxidant-induced lipid peroxidation in rat brain in vitro (IC_50_ value of 8.01 mg/mL and 6.3 mg/mL, respectively) [49]. At 5 mg/kg 3,5-dicaffeoylquinic acid reduced significantly the blood glucose levels and ameliorate the oxidative stress biomarkers reduced glutathione, malondialdehyde and serum biochemical parameters [50].

### 2.6. PLS-DA Analysis

Based on the antioxidant, enzyme inhibitory, a supervised partial least square discriminant analysis (PLS-DA) was simulated considering parts as class membership criteria, and the outcomes are summarized in Figure 2.

The discriminant analysis resulted in a good segregation of the three parts (Figure 2A). Figure 2B. showed the performance of the model evaluated through the Area Under the Curve (AUC) average using one-vs-all comparisons. An AUC value of 1 was obtained when taking account 2 function, suggesting the great segregation between the three parts along the first two function. By referring to Figure 2C, it appears that the first function separated the samples based on DPPH, ABTS, CUPRAC, FRAP, MCA, AChE, BChE, tyrosinase and amylase activities, while the second function separated the samples according to PBD, glucosidase and lipase activities. Regarding Figure 2D, the strongest activity recorded by the flower heads extract were DDPH, FRAP, CUPRAC and ABTS. Similarly, the roots proved to be the most effective plant part to give better PBD, anti-glucosidase and anti-lipase properties, while the higher anti-BChE, anti-AChE and anti-tyrosinase activity were recorded by the leaves extract. Furthermore, the contribution of the metabolites in the biological activities was evaluated through the Pearson’s correlation analysis. As reported in Appendix A, several metabolites seem to be involved in various biological activities, since a positive Pearson coefficient higher than 0.7 was obtained. Some metabolites are well known in the literature for their various properties; protocatechuic acid, syringic acid, isorhamnetin and quercetin have been reported for potential action, such as antioxidant activity [51,52,53,54]. In addition, neochlorogenic was reported to be the predominant antioxidant compound in *Polygonum cuspidatum* leaves [55]. Further, experimental studies support the effectiveness of protocatechuic acid and vanillic acid in the prevention of diabetes diseases and neurodegenerative processes, including Alzheimer’s [56,57]. In addition, *p*-coumaric acid is well known for its antioxidant activity, prevention and improvement of diabetes and neuroprotection [58].

### 2.7. Cytotoxicity Assay

To assess the cytotoxicity of the extracts, we used the common THP-1 cells, a human monocytic cell line that mimics the behavior of the costmary extracts towards the immune system (Figure 3). After 24 h of incubation of macrophage cell line THP-1 with flower heads, roots, and leaves extracts, we observe 25% toxicity for the three extracts at concentration of 2 µg mL^−1^. A toxicity of 50% was reached at 200 µg mL^−1^ for flower heads extract, 1000 µg mL^−1^ and 2000 µg mL^−1^ for leaves and roots extracts, respectively, which may be regarded as a very high concentration, without any biological meaning. At a concentration of 3000 µg mL^−1^, a 100% toxicity for flower heads extract was observed.

## 3. Materials and Methods

### 3.1. Plant Material

The leaves, flower heads and roots of *T. balsamita* were collected from herbal garden (Belopoptsi village, Gorna Malina region) in Bulgaria at 700 m a.s.l. (42.67° N 23.77° E), during the full flowering stage in July 2021. The seedlings were provided by the greenhouse “Zelena prolet” (Sofia, Bulgaria). The plant was identified by one of us (V. B.) according to Kuzmanov [59]. The voucher specimen was deposited at Herbarium Academiae Scientiarum Bulgariae (SOM 177 806). Seven plant samples were separate into roots, leaves and flower heads and dried at room temperature.

### 3.2. Sample Extraction

Air-dried powdered leaves, roots and flower heads (10 g) were extracted with 80% MeOH (1:20 *w*/*v*) by sonication (80 kHz, ultrasound bath Biobase UC-20C) for 15 min (×2) at room temperature. The extracts were concentrated in vacuo and lyophilized (lyophilizer Biobase BK-FD10P) (Jinan, China) to yield crude extracts as follows: flower heads 1.08 g, leaves 1.93 g and roots 0.68 g. The lyophilized extracts were dissolved in 80% methanol (0.1 mg/mL). An aliquot (2 mL) of each extract solution was filtered through a 0.45 μm syringe filter (Polypure II, Alltech, Lokeren, Belgium) and subjected to UHPLC–HRMS analyses.

### 3.3. Chemicals

Acetonitrile (hypergrade for LC–MS), formic acid (for LC-MS) and methanol (analytical grade) were purchased from Merck (Merck, Bulgaria). The authentic standards used for compound identification were obtained from Extrasynthese (Genay, France) for protocatechuic, syringic, vanillic gentisic, *p*-coumaric, *m*-coumaric, *o*-coumaric acid, quercetin, apigenin, luteolin, apigenin 7-*O*-glucoside, kaempferol 3-*O*-glucoside, isorhamnetin 3-*O*-glucoside, luteolin 7-*O*-glucoside and rutin, and kaempferol 3-*O*-rutinoside, homoorientin, isoquercitrin, hyperoside, nepetin. Caffeic acid, neochlorogenic acid, 3,4-dicaffeoylquinic acid, 1,5-dicaffeoylquinic acid, jaceosidin and hispidulin were supplied from Phytolab (Vestenbergsgreuth, Germany). Chlorogenic acid, isorhamnetin and cirsimaritin were purchased from Sigma-Aldrich (St. Louis, MO, USA).

### 3.4. Ultra High-Performance Liquid Chromatography—High Resolution Mass Spectrometry (UHPLC—HRMS)

Mass spectrometry analyses were carried out on a Q Exactive Plus mass spectrometer (ThermoFisher Scientific, Inc., Waltham, MA, USA) equipped with a heated electrospray ionization (HESI-II) probe (ThermoScientific). The mass spectrometer was operated in negative and positive ESI modes within the *m*/*z* range from 100 to 1000. The other parameters were as follows: spray voltage 3.5 kV (+) and 2.5 kV (−); sheath gas flow rate 38; auxiliary gas flow rate 12; spare gas flow rate 0; capillary temperature 320 °C; probe heater temperature 320 °C; S-lens RF level 50 and scan mode: full MS (resolution 70,000) and MS/MS (17,500). The chromatographic separation was performed on a reversed-phase column Kromasil EternityXT C18 (1.8 µm, 2.1 × 100 mm) at 40 °C. The chromatographic analyses were run using 0.1% formic acid in water (A) and 0.1% formic acid in acetonitrile (B) as the mobile phases. The flow rate was 0.3 mL/min. The run time was 33 min. The following gradient elution program was used: 0–1 min, 0–5% B; 1–20 min, 5–30% B; 20–25 min, 30–50% B; 25–30 min, 50–70% B; 30–33 min, 70–95% and 33–34 min 95–5% B. Equilibration time was 4 min [21]. Data were processed by Xcalibur 4.2 (ThermoScientific) instrument control/data handling software. Metabolite profiling using MZmine 2 software was applied to the UHPLC–HRMS raw files of the studied *T. balsamita* extracts. The areas under the curve (AUC) for each identified/annotated compound were plotted and used for further statistical analysis in 3.10.

### 3.5. Data Filtering for Annotation of Target Compounds

Raw (ThermoFisher Scientific) mass spectrometric files were converted to .ms1 (MS1 data) and .mgf (MS2 data) by MSConvertGUI 3.1 (ProteoWizard) and manipulated under the R programming language (version 4.2.1, 2022-06-23, Funny-Looking Kid). The MS2 spectra were screened for the presence of the available target (hydroxybenzoic acid derivatives and flavonoids) compounds. The screening was achieved by selecting spectra based on the following criteria: *m*/*z* error of the molecular ion <15 ppm, retention time error <2%, number of fragment ions match >2/3, absolute error of the percentage intensity of matched fragment ions <15. Spectra identified as the same reference compound found in the same chromatographic peak were grouped, i.e., the spectra were summed, the *m*/*z* were adjusted by weight averaging where is the recalculated *m*/*z* value and *int*_i_ are the *m*/*z* and the intensity of the ith fragment ion, respectively.

### 3.6. Total Phenolic and Flavonoid Contents

Total phenols and flavonoids were measured as gallic acid (GAE) and rutin (RE) equivalents respectively, through validated spectrophotometric methods. The experiments were carried out as reported in previous studies [60,61,62]. The detailed protocols are given in Appendix A.

### 3.7. Determination of Antioxidant and Enzyme Inhibitory Effects

Intrinsic scavenging/reducing properties of the extracts (0.2–1 mg/mL) were determined through colorimetric assays [21]. Additionally, extracts (0.2–1 mg/mL) were assayed for evaluating enzyme inhibition effects towards tyrosinase, α-amylase, α-glucosidase and cholinesterases and lipase. Detailed protocols were reported elsewhere [21,62,63,64]. The detailed antioxidant and enzyme inhibitory assays are given in Appendix A.

### 3.8. Cell Line and Culture

The human monocytic THP-1 (TIB-202) cell line was obtained from the American Type Culture Collection (ATCC, Manassas, VA, USA). The cells grown in RPMI 1640 were supplemented with 10% fetal bovine serum, 1% glutamine, 1% penicillin/streptomycin and 0.5% Amphotericin B. Cells were cultured in a humidified atmosphere at 37 °C under a 5% CO_2_ atmosphere.

### 3.9. Cytotoxicity Assay

THP-1 cells (at 1.10^5^ cells/mL) in RPMI medium (Thermo-Fisher) supplemented with 10% FBS were seeded in each well of a 48-well plate (*n* = 4). Cells were permitted to adhere for 24 h, and then treated with roots, flowers and leaves extracts of *T. balsamita* in a medium for 24 h. Then, 40 µL of WST-1 testing solutions (Sigma-Aldrich) was added to each well and the plate incubated at 37 °C for 2 h. The contents of each well were laid down in 3 wells of a 96-well plate [65]. The absorbances were measured at 350 and 630 nm with an Omega StarLab spectrophotometer (Omega, Ortenberg, Germany).

### 3.10. Statistical Analysis

In the antioxidant and enzyme inhibitory assays, the values are expressed as mean ± SD of three parallel experiments.

In terms of antioxidant and enzyme inhibitory abilities, one way ANOVA with Tukey’s assay was performed to determine differences between the tested extracts. The statistical analysis was performed using XlStat 16.0 software. Clustered Image Maps (CIM) were used to visualize metabolite variation among the extracts. Prior to CIM analyses, data were normalized and centered. Afterwards, a supervised Partial Least-Square discriminant analysis (PLS-DA) was done to discriminate the different parts regarding their biological activities. Then, CIM was applied on PLS-DA outcomes to characterize each extract. Lastly, Pearson’s correlation coefficients were calculated to evaluate the relationship between secondary metabolites and the biological activities, respectively.

## 4. Conclusions

More than 100 secondary metabolites, including methoxylated flavonols and flavones, acylquinic acids analogues, hydroxybenzoic and hydroxycinnamic acids derivatives, and their glycosides, were annotated/dereplicated in the costmary leaves, flower heads and roots extracts. Ninety-one compounds are reported in the species for the first time. Chlorogenic, 3, 5-*di*CQA and 4, 5-*di*CQA acid dominated the leaves and roots extracts profiles. Despite the previously published data on the high concentration of cichoric acid in the costmary aerial parts extract, we were not able to confirm the presence of either cichoric acid or any esters of tartaric acid and hydroxycinnamic acid. According to this study, the presence of 6-methoxylated flavones and flavonols, dicaffeoylquinic acids and their hexosides and phenolic acids glycosides could be considered significant in the chemotaxonomy of the *Tanacetum* genus. To understand the relationship between plant parts and biological activity, multivariate statistical analyses were performed. The strongest antioxidant activity (DDPH, FRAP, CUPRAC and ABTS) of the flower heads extract could be related to the presence of rutin, isoquercitrin and hyperoside, and the corresponding aglycone. A variety of acylquinic acids, flavoneshexuronides and methoxylated aglycones in the leaves extract could be associated with its anti-BChE, anti-AChE and anti-tyrosinase activity. Phenolic acidshexosides, di- and tri- caffeoylquinic acids accounted for the stronger α-glucosidase and α-lipase inhibitory activity of the roots extracts. The assayed extracts expressed low cytotoxicity towards THP-1 viability. In addition to evoking an antioxidant response, costmary extracts display in vitro enzyme inhibitory effects, which generate interest in the plant as a valuable herbal drug. Moreover, this study advocates further work geared towards additional in vivo studies.

## Figures and Tables

**Figure 1 plants-12-00022-f001:**
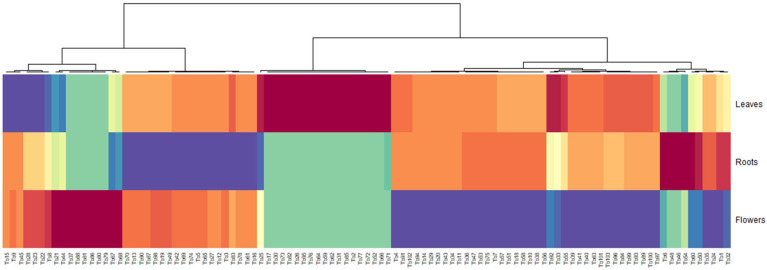
Global overview of the metabolite’s contrasts among *Tanacetum balsamita* samples (Clustered Image Map).

**Figure 2 plants-12-00022-f002:**
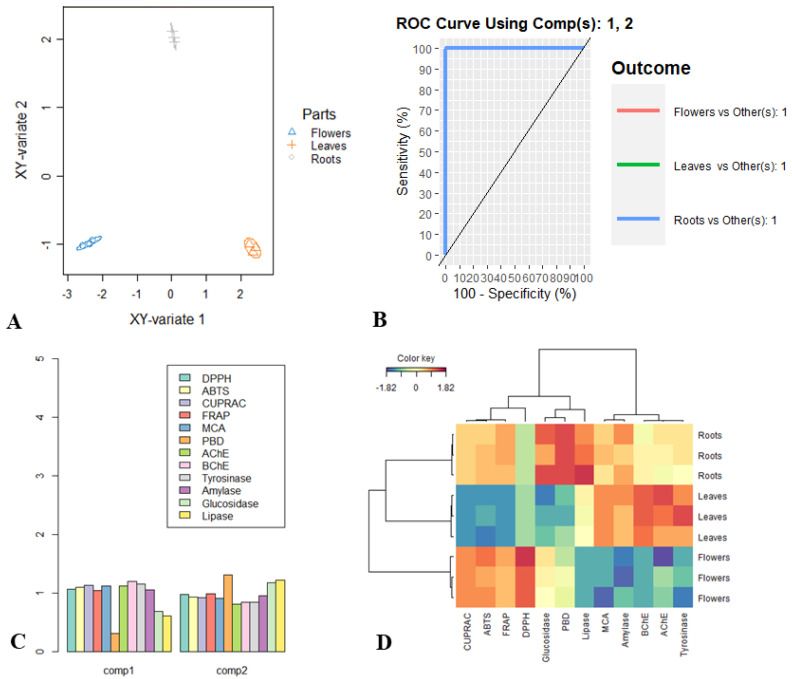
Partial Least-Squares Discriminant analysis graphical outputs on the biological activities of *Tanacetum balsamita.* (**A**) Samples plot. (**B**) ROC curve and AUC averaged using one-vs.-all comparisons. (**C**) The most discriminant biological activities identifying though VIP score calculation. (**D**) Clustered Image map (Ward linkage, Euclidean distance).

**Figure 3 plants-12-00022-f003:**
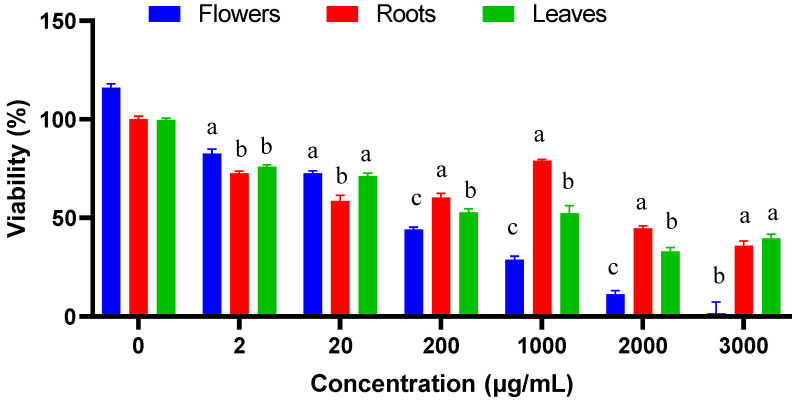
Cytotoxicity of *T. balsamita* extracts towards THP-1 cells (one-way ANOVA, different letters indicate significant difference between plant parts in the same concentration (a, b and c), *p* ≤ 0.05).

**Table 1 plants-12-00022-t001:** Total phenolic and flavonoid contents of the *T. balsamita* extracts.

Samples	TPC (mg GAE/g)	TFC (mg QE/g)
Leaves	30.82 ± 0.16 ^c^	18.97 ± 0.44 ^b^
Roots	43.41 ± 0.30 ^b^	3.74 ± 0.07 ^c^
Flower heads	59.75 ± 0.66 ^a^	41.02 ± 0.50 ^a^

Values are reported as mean ± SD of three parallel experiments. GAE: Gallic acid equivalent; QE: Quercetin equivalents; TE: Trolox equivalent. Different letters indicate significant differences in the tested extracts (*p* < 0.05).

**Table 2 plants-12-00022-t002:** Secondary metabolites in *Tanacetum balsamita* extracts assayed by UHPLC-ESI-MS/MS.

No	Identified/Tentatively Annotated Compound	Molecular Formula	Exact Mass [M-H]^−^	Fragmentation Pattern in (-) ESI-MS/MS	t_R_ (min)	Δ ppm	Distribution
**Hydroxybenzoic, Hydroxycinnamic and Acylquinic Acids, and Derivatives**							
**1**	protocatechuic acid-*O*-hexoside ^b^	C_13_H_16_O_9_	315.0727	315.0724 (100), 153.0180 (26.8), 152.0101 (61.7), 123.0071 (3.4), 109.0287 (9.5), 108.0200 (92.3)	1.72	0.840	1,2,3
**2**	hydroxybenzoic acid-pentosylhexoside^b^	C_18_H_23_O_12_	431.1198	431.1198 (63.96), 137.0230 (100), 93.0329 (77.34)	1.83	0.582	2
**3**	protocatechuic acid ^a,b^	C_7_H_6_O_4_	153.0182	153.0180 (17.8), 125.0228 (0.4), 109.0279 (100), 91.0174 (1.7), 81.0329 (1.8)	2.05	−8.574	1,2,3
**4**	protocatechuic acid-*O*-hexoside isomer ^b^	C_13_H_16_O_9_	315.0753	315.0729 (37.5), 153.0544 (100), 123.0436 (54.5), 109.0279 (34.8)	2.15	0.840	1,2,3
**5**	*p*-hydroxyphenylacetic acid 1-*O*-hexoside ^b^	C_14_H_18_O_8_	313.0727	313.0941 (1.1), 151.0387 (100), 121.0281 (3.4), 107.0486 (98.6)	2.17	4.022	1,2
**6**	syringic acid ^a^	C_9_H_10_O_5_	197.0455	197.0444 (25.1), 182.0210 (100), 166.9974 (26.1), 153.0547 (5.3), 138.0309 (16.7), 123.0072 (46.0), 95.0122 (15.4)	2.29	−5.819	1,2,3
**7**	syringic acid 4-*O*-hexoside ^b^	C_15_H_20_O_10_	359.0984	359.0986 (8.2), 197.0446 (100), 182.0210 (18.1), 153.0543 (14.5), 138.0308 (27.2), 123.0072 (29.5)	2.30	0.362	2,3
**8**	neochlorogenic (3-caffeoylquinic) acid ^a,b^	C_16_H_18_O_9_	353.0867	353.0879 (43.1), 191.0550 (100), 179.0338 (62.8), 173.0442 (3.2), 161.0232 (3.5), 135.0437 (53.1), 93.0330 (4.9), 85.0278 (9.3)	2.38	0.240	1,2,3
**9**	caffeic acid- *O*-hexoside ^b^	C_15_H_18_O_9_	341.0867	341.0867 (4.27), 179.0338 (100), 135.0436 (0.91), 107.0485 (0.91)	2.42	−3.153	1,2,3
**10**	vanillyl-*O*-hexose ^b^	C_14_H_18_O_9_	329.0875	329.0875 (100), 329.0674 (5.0), 209.0448 (32.0), 167.0338 (46.7), 152.0105 (4.8), 123.0433 (4.3)	2.50	−1.049	2,3
**11**	gentisic acid-O-hexoside ^b^	C_13_H_16_O_9_	315.0727	315.0723 (38.1), 153.0183 (68.4), 135.0071 (3.8), 109.0279 (100), 91.0174 (0.5), 65.0380 (6.4)	2.58	0.555	2,3
**12**	aesculetin-*O*-hexoside ^b^	C_15_H_15_O_9_	339.0724	339.0723 (24.7), 177.0182 (100), 149.0227 (1.4), 133.0280 (10), 105.0330 (3.9), 89.0381 (1.9)	2.69	0.781	1,2
**13**	vanillic acid ^a,b^	C_8_H_8_O_4_	167.0350	167.0337 (100), 137.0230 (4.9), 123.0437 (21.8), 152.0092 (0.2), 108.0201 (100), 95.0486 (1.9)	3.03	−7.735	1,2
**14**	caffeoylgluconic acid ^b^	C_15_H_18_O_10_	357.0827	357.0835 (16.6), 195.0501 (100), 179.0339 (45.0), 177.0394 (11.2), 165.0398 (0.5), 147.0284 (4.8), 135.0437 (42.24), 129.0180 (6.1), 87.0072 (8.7), 59.0123 (1.9)	2.82	2.185	2,3
**15**	*O*-caffeoyl hexose ^b^	C_15_H_18_O_9_	341.0867	341.0875 (20.5), 281.0665 (77.4), 251.0557 (37.6), 221.0449 (31.7), 179.0338 (100), 161.0231 (45.3), 135.0437 (60.1), 111.0438 (8.7)	2.83	−1.012	1,2,3
**16**	4-hydroxybenzoic acid ^a,b^	C_7_H_6_O_3_	137.0230	137.0230 (100), 119.0126 (1.8), 108.0202 (7.3), 93.0330 (3.7), 65.0380 (0.9)	2.86	−10.052	1,2,3
**17**	*p*-hydroxyphenylacetic acid-*O*-hexoside isomer ^b^	C_14_H_18_O_8_	313.0936	313.0934 (13.6), 151.0387 (100), 123.0070 (1.1), 109.0281 (2.7)	3.01	1.754	2
**18**	hydroxybenzoic acid-*O*-hexoside ^b^	C_13_H_16_O_8_	299.0778	299.0779 (1.6), 137.0230 (100), 93.0330 (53.4)	3.02	2.238	2,3
**19**	dihydroxyphenylacetic acid-*O*-pentosylhexoside ^b^	C_22_H_21_O_11_	461.1115	461.1115 (10.80), 281.0454 (15.56), 167.0337 (100), 149.0230 (82.64), 123.6436 (73.22), 108.0199 (29.20), 95.0486 (7.49)	3.03	5.520	1,2
**20**	caffeoylgluconic acid isomer ^b^	C_15_H_18_O_10_	357.0827	357.0830 (1.7), 195.0502 (100), 179.0338 (6.6), 177.0398 (5.1), 161.0230 (4.0), 135.0437 (10.5), 129.0179 (14.8), 87.0072 (5.2), 59.0123 (3.4)	3.07	0.812	2,3
**21**	caffeic acid-*O*-hexoside isomer ^b^	C_15_H_18_O_9_	341.0867	341.0879 (27.9), 179.0338 (100), 135.0437 (71.2), 107.0487 (1.0)	3.12	0.160	1,2,3
**22**	quinic acid ^b^	C_7_H_12_O_6_	191.0561	191.0550 (100), 173.0446 (2.0), 155.0337 (0.3), 127.0386 (3.8), 111.0436 (1.8), 93.0330 (5.9), 85.0278 (19.8)	3.19	−5.921	1,2,3
**23**	chlorogenic (5-caffeoylquinic) acid ^a^	C_16_H_18_O_9_	353.0867	353.0880 (5.1), 191.0550 (100), 173.0444 (1.4), 161.0232 (1.7), 127.0385 (1.9), 111.0435 (1.0), 93.0330 (3.3), 85.0278 (8.7)	3.19	0.495	1,2,3
**24**	caffeic acid-*O*-hexoside ^b^	C_15_H_18_O_9_	341.0867	341.0878 (10.4), 179.0338 (100), 135.0436 (73.3), 107.0488 (0.8)	3.27	−0.104	1,2,3
**25**	coumaric acid-*O*-hexoside ^b^	C_15_H_18_O_8_	325.0930	325.0930 (6.1), 163.0385 (53.5), 135.0435 (0.4), 119.0486 (100)	3.32	0.305	1,2,3
**26**	*p*-hydroxyphenylacetic acid-*O*-hexoside isomer ^b^	C_14_H_18_O_8_	313.0934	313.0932 (5.1), 151.0386 (10.2), 107.0486 (100)	3.31	0.988	2
**27**	*p*-coumaric acid ^a^	C_9_H_8_O_3_	163.0389	163.0387 (3.4), 135.0072 (1.1), 119.0486 (100)	3.35	−8.510	1,2
**28**	4-caffeoylquinic acid ^a,b^	C_16_H_18_O_9_	353.0867	353.0880 (31.1), 191.0554 (45.9), 179.0341 (68.1), 173.0446 (100), 135.0439 (52.6), 111.0437 (2.8), 93.0332 (20.7), 85.0280 (8.0)	3.36	0.551	1,2,3
**29**	caffeoylgluconic acid isomer ^b^	C_15_H_18_O_10_	357.0827	357.0820 (5.0), 195.0652 (100), 179.0541 (0.2), 177.0410 (1.0), 135.0437 (601), 59.0123 (11.6)	3.41	−2.100	2,3
**30**	3-feruloylquinic acid ^b^	C_17_H_20_O_9_	367.1035	367.1028 (22.3), 193.0496 (100), 191.0556 (2.5), 173.0443 (4.5), 134.0358 (48.1), 127.0389 (0.5), 93.0329 (1.8)	3.43	−1.921	2
**31**	*p*-hydroxyphenylacetic acid ^a,b^	C_8_H_8_O_3_	151.0401	151.0386 (100), 107.0486 (0.59), 136.0154 (0.48), 123.0072 (4.00)	3.47	−9.715	2
**32**	caffeic acid ^a^	C_9_H_8_O_4_	179.0338	179.0339 (21.1), 135.0436 (100), 117.0330 (0.6), 107.0487 (1.3)	3.54	−6.211	1,2,3
**33**	gentisic acid ^a^	C_7_H_6_O_4_	153.0182	153.0180 (84.5), 135.0073 (32.7), 125.0233 (0.4), 109.0279 (100), 91.0173 (6.1), 81.0331 (0.4), 65.0380 (18.9)	3.84	−8.901	2,3
**34**	5-*p*-coumaroylquinic acid ^b^	C_16_H_18_O_8_	337.0929	337.0932 (9.3), 191.0549 (100), 173.0444 (7.1), 163.0388 (6.6), 119.0487 (5.3), 111.0436 (2.9), 93.0329 (17.5), 85.0278 (4.7)	3.96	1.096	2,3
**35**	3-hydroxy-dihydrocaffeoyl-5-caffeoylquinic acid ^b^	C_25_H_26_O_13_	533.1301	533.1298 (100), 371.0992 (19.1), 353.0880 (16.7), 191.0551 (84.2), 179.0339 (66.7), 161.0236 (4.9), 135.0437 (88.2), 93.0329 (15.5)	4.05	−0.570	2,3
**36**	5-feruloylquinic acid ^b^	C_17_H_20_O_9_	367.1035	367.1035 (18.8), 191.0550 (100), 173.0443 (11.8), 155.0336 (0.5), 134.0360 (9.0), 111.0435 (4.1), 93.0329 (22.1)	4.41	−0.015	2,3
**37**	dihydroxyphenylacetic acid ^b^	C_8_H_8_O_4_	167.0341	167.0344 (1.3), 137.0230 (2.0), 123.0436 (19.3), 108.0200 (100)	4.41	−5.520	1,2,3
**38**	1-caffeoyl-3-hydroxy-dihydrocaffeoylquinic acid ^b^	C_25_H_26_O_13_	533.1301	533.1313 (27.6), 371.0985 (49.1), 353.0902 (4.3), 335.0750(2.0), 191.0551 (13.4), 179.0342 (11.8), 173.0444 (23.0), 161.0232 (2.5), 135.0436 (100), 111.0436 (1.6), 93.0330 (8.0)	4.45	2.412	2,3
**39**	coumaric acid-*O*-hexoside isomer ^b^	C_15_H_18_O_8_	325.0931	325.0930 (1.6), 163.0387 (100), 119.0487 (98.8)	4.45	0.398	2,3
**40**	*m*-coumaric acid ^a,b^	C_9_H_8_O_3_	163.0389	163.0387(2.86), 135.0434 (11.34), 119.0487 (100)	4.46	−7.651	2,3
**41**	*o*-coumaric acid ^a,b^	C_9_H_8_O_3_	163.0389	163.0387 (170), 135.0436 (11.3), 119.0487 (100)	4.56	−8.142	2,3
**42**	5-*p*-coumaroylquinic acid isomer ^b^	C_16_H_18_O_8_	337.0929	337.0932 (7.8), 191.0550 (100), 173.0444 (2.8), 163.0388 (1.9), 127.0385 (1.7), 119.0486 (1.5), 111.0433 (1.3), 93.0329 (5.2), 85.0278 (7.2)	4.62	0.829	1,2
**43**	4-feruloylquinic acid ^b^	C_17_H_20_O_9_	367.1035	367.1035 (96.7), 193.0496 (11.3), 191.0552 (0.7), 173.0446 (70.1), 134.0358 (24.5), 111.0435 (15.6), 93.0329 (100)	4.68	0.122	2,3
**44**	3,5-dicaffeoylquinic acid-hexoside ^b^	C_31_H_34_O_17_	677.1512	677.1538 (53.28), 515.1409 (100), 353.0878 (7.1), 341.0879 (14.5), 323.0774 (56.3), 335.0778 (4.1), 191.0551 (99.7), 179.0340 (44.8), 173.0446 (6.1), 161.0231 (44.2), 135.0437 (42.5), 127.0382 (2.1), 93.0329 (10.4)	5.16	3.850	1,2,3
**45**	4,5-dicaffeoylquinic acid-hexoside ^b^	C_31_H_34_O_17_	677.1512	677.1729 (100), 515.1287 (15.2), 353.0862 (24.8), 341.0890 (2.9), 323.0792 (19.4), 191.0553 (33.6), 179.0340 (60.5), 173.0443 (71.6), 161.0232 (26.3), 135.0438 (65.6), 93.0328 (17.1)	5.56	0.779	1,2,3
**46**	3,4-dicaffeoylquinic acid ^a^	C_25_H_24_O_12_	515.1195	515.1198 (100), 353.0880 (14.3), 335.0774 (5.9), 203.0340 (0.8), 191.0551 (29.5), 179.0339 (50.0), 173.0444 (62.9), 161.0230 (16.4), 135.0437 (50.0), 111.0436 (4.4), 93.0329 (15.6)	5.70	0.487	1,2,3
**47**	3-dehydrocaffeoyl-5-caffeoylquinic acid ^b^	C_25_H_22_O_12_	513.1038	513.1042 (61.9), 351.0724 (100), 335.0770 (10.0), 191.0551 (18.5), 179.0339 (42.8), 177.0182 (53.8), 173.0443 (35.9), 161.0231 (15.7), 135.0434 (47.9), 133.0280 (86.1), 93.0329 (18.9)	5.85	0.898	2, 3
**48**	3,5-dicaffeoylquinic acid ^a^	C_25_H_24_O_12_	515.1195	515.1199 (19.8), 353.0877 (98.3), 335.0765 (0.5), 191.0550 (100), 179.0338 (49.6), 173.0445 (3.6), 161.0232 (4.3), 135.0436 (49.1), 127.0385 (2.4), 111.0433 (1.7), 93.0330 (3.7), 85.0278 (7.3)	5.87	0.137	1,2,3
**49**	dihydroxyphenylacetic acid-*O*-dipentosyl-hexoside ^b^	C_27_H_29_O_15_	593.1543	593.1543 (2.33), 461.1083 (0.84), 167.0338 (100), 149.0230 (5.91), 131.0699 (9.05), 123.0430 (39.75), 108.0200 (90.10)	6.19	5.238	1,2
**50**	4,5-dicaffeoylquinic acid ^b^	C_25_H_24_O_12_	515.1195	515.1198 (92.5), 353.0879 (54.9), 335.0771 (0.9), 191.0550 (36.1), 179.0337 (65.6), 173.0442 (100), 161.0230 (5.3), 135.0435 (64.3), 111.0435 (4.0), 93.0328 (25.2)	6.23	0.390	1,2,3
**51**	shikimic acid ^b^	C_7_H_10_O_5_	173.0455	173.0443 (100), 155.0337 (1.6), 127.0381 (1.3), 111.0434 (9.3), 93.0329 (61.6)	6.22	−7.147	2,3
**52**	rosmarinic acid ^a^	C_18_H_16_O_8_	359.0778	359.0778 (16.3), 197.0447 (29.2), 179.0341 (12.8), 161.0231 (100), 135.0437 (16.0)	6.33	1.781	2
**53**	3-feruloyl-4-caffeoylquinic acid ^b^	C_26_H_26_O_12_	529.1351	529.1352 (100), 367.1038 (3.8), 353.0878 (6.2), 335.0771 (11.3), 193.0496 (52.2), 191.0552 (7.1), 179.0340 (35.7), 173.0444 (39.5), 161.0233 (20.3), 135.0439 (30.8), 134.0359 (38.6), 111.0436 (5.9), 93.0331 (10.3)	6.50	0.096	2,3
**54**	3-*p*-coumaroyl-5-caffeoylquinic acid ^b^	C_25_H_24_O_11_	499.1246	499.1254 (31.7), 353.0872 (0.4), 337.0931 (75.6), 335.0769 (2.1), 191.0550 (9.7), 173.0443 (8.4), 163.0388 (100.0), 135.0437 (2.9), 119.0487 (37.5), 93.0330 (4.1)	6.52	1.694	1,2,3
**55**	1-*p*-coumaroyl-5-caffeoylquinic acid ^b^	C_25_H_24_O_11_	499.1246	499.1235 (36.8), 353.0880 (45.8), 337.0934 (59.7), 191.0551 (100), 179.0337 (33.7), 173.0444 (18.6), 163.0388 (49.4), 135.0436 (39.6), 119.0484 (25.8)	6.80	−2.173	1,2,3
**56**	3-feruloyl-5-caffeoylquinic acid ^b^	C_26_H_26_O_12_	529.1351	529.1354 (54.2), 367.1034 (97.1), 335.0782 (2.3), 193.0497 (100), 191.0546 (11.6), 173.0443 (53.4), 161.0230 (22.3), 135.0441 (10.8), 134.0358 (86.4), 111.0437 (3.3), 93.0330 (13.9)	6.82	0.190	2,3
**57**	4-feruloyl-5-caffeoyl quinic acid ^b^	C_26_H_26_O_12_	529.1351	529.1354 (92.8), 367.1034 (100), 353.0876 (5.6), 193.0496 (8.9), 191.0546 (10.5), 179.0338 (44.9), 173.0444 (65.7), 161.0231 (20.4), 135.0437 (56.2), 134.0358 (22.6), 111.0437 (11.2), 93.0329 (75.0)	7.02	0.549	2,3
**58**	4-caffeoyl-5-feruloylquinic acid ^b^	C_26_H_26_O_12_	529.1351	529.1359 (7.2), 367.1042 (12.0), 353.0875 (49.7), 193.0486 (1.4), 191.0551 (58.1), 179.0337 (61.9), 173.0444 (83.7), 161.0230 (21.8), 135.0437 (65.7), 134.0360 (2.6), 111.0436 (5.0), 93.0330 (30.6)	7.18	1.343	2,3
**59**	4-caffeoyl-5-*p*-coumaroylquinic acid ^b^	C_25_H_24_O_11_	499.1246	499.1253 97.66), 353.0868 (76.7), 337.0949 (7.9), 191.0549 (73.8), 179.0338 (75.5), 173.0442 (100), 161.0233 (7.6), 135.0437 (89.6), 111.0437 (8.9), 93.0329 (29.0)	7.63	1.453	2
**60**	3,4,5-tricaffeoylquinic acid ^b^	C_34_H_30_O_15_	677.1512	677.1517 (94.2). 515.1199 (31.6), 353.0879 (55.7), 335.0774 (14.1), 299.0594 (1.3), 255.0676 (1.7), 203.0349 (3.9), 191.0551 (47.7), 179.0338 (76.8), 173.0443 (100), 161.0232 (28.7), 135.0436 (82.0), 111.0435 (5.6), 93.0330 (24.3)	7.78	0.748	1,2,3
**Flavonoids**							
**61**	naringenin 6, 8 di*C*-hexoside ^b^	C_27_H_32_O_15_	595.1678	595.1680 (100), 475.1255 (3.8), 457.1151 (2.5), 415.1039 (11.2), 385.0930 (30.4), 355.0826 (37.6), 271.0618 (0.6), 163.0027 (1.4), 151.0017 (1.0), 119.0487 (15.2), 107.0123 (3.3)	3.64	1.994	1,2
**62**	apigenin 6, 8-di*C*-hexoside ^b^	C_27_H_29_O_15_	593.1512	593.1518 (100), 503.1208 (4.7), 473.1090 (16.0), 413.0892 (2.0), 395.0779 (1.9), 383.0775 (18.6), 353.0669 (32.6), 325.0706 (2.4), 297.0767 (10.9), 161.0233 (2.0), 117.0329 (3.2)	4.04	0.905	2
**63**	homoorientin (luteolin 6-*C*-glucoside) ^a,b^	C_21_H_20_O_11_	447.0933	447.0930 (100), 369.0610 (2.5), 357.0614 (39.3), 339.0497 (2.4), 327.0514 (53.7), 311.0537 (1.7), 299.0573 (3.5), 298.0487 (3.3), 297.0411 (14.0), 285.0405 (3.8), 133.0280 (11.4), 175.0376 (2.9)	4.54	0.225	1, 2,3
**64**	luteolin *O*-hexuronosyl-*O*-hexoside ^b^	C_27_H_28_O_17_	623.1264	623.1263 (66.0), 447.0930 (2.6), 285.0403 (100), 257.0454 (0.5), 243.0290 (0.9), 217.0499 (1.5), 199.0393 (2.5), 175.0391 (2.5), 151.0025 (3.9), 133.0280 (7.5), 107.0125 (2.6)	4.72	1.457	2
**65**	rutin ^a^	C_27_H_30_O_16_	609.1464	609.1467 (100), 301.0346 (30.2), 300.0274 (79.6), 271.0247 (39.9), 255.0296 (17.3), 243.0294 (8.7), 227.0345 (2.3), 211.0391 (0.4), 178.9976 (2.7), 163.0022 (1.4), 151.0023 (5.5), 121.0277 (0.3), 107.0121 (2.3)	5.08	0.972	1,2
**66**	luteolin *O*-pentosylhexoside ^b^	C_26_H_28_O_15_	579.1360	579.1364 (83.4), 447.0879 (0.5), 285.0404 (100), 256.0366 (1.3), 241.0502 (0.6), 227.0341 (0.8), 175.0385 (2.0), 151.0024 (5.0), 133.0280 (3.7), 107.0124 (2.3)	5.09	1.394	2
**67**	isoquercitrin ^a^	C_21_H_20_O_12_	463.0886	463.0887 (100), 343.0472 (0.5), 301.0346 (37.6), 300.0274 (82.0), 271.0248 (32.5), 255.0296 (13.3), 243.0296 (8.2), 227.0344 (2.6), 211.0398 (0.5), 178.9979 (2.30), 163.0033 (1.5), 151.0024 (9.1), 121.0275 (1.0), 107.0124 (3.8)	5.18	1.103	1,2,3
**68**	hyperoside ^a,b^	C_21_H_20_O_12_	463.0887	463.0829 (100), 301.0352 (12.6), 300.0272 (21.7), 271.0245 (10.2), 255.0283 (4.4), 243.0284 (2.2), 179.0331 (1.8), 175.0245 (8.0), 163.0372 (1.5), 151.0023 (50.4), 135.0438 (40.4), 107.0123 (12.3)	5.29	1.218	1,2,3
**69**	nepetin *O*-pentosylhexoside ^b^	C_27_H_30_O_16_	609.1468	609.1468 (100), 315.0516 (63.35), 301.0354 (7.20), 300.0279 (32.54), 299.0202 (8.60), 285.0401 (5.50), 271.0251 (1.33), 133.0282 (6.77)	5.35	−5.123	1,2
**70**	luteolin 7-*O*-rutinoside ^b^	C_27_H_30_O_15_	593.1512	593.1518 (83.0), 285.0403 (100), 256.0372 (0.6), 243.0290 (0.6), 229.0499 (0.4), 217.0492 (0.8), 175.0391 (2.2), 151.0023 (4.7), 133.0281 (4.6), 107.0119 (1.8)	5.22	1.006	1,2
**71**	luteolin 7-*O*-glucoside ^a^	C_21_H_20_O_11_	447.0933	447.0935 (100), 285.0404 (43.0), 284.0324 (49.0), 255.0291 (0.7), 227.0349 (3.2), 211.0394 (2.7), 161.0230 (1.9), 151.0025 (3.8), 133.0280 (4.1), 107.0122 (2.3)	5.31	0.437	1, 2,3
**72**	luteolin *O*-hexuronide ^b^	C_21_H_18_O_12_	461.0736	461.0730 (54.1), 285.0403 (100), 267.0295 (0.3), 243.0297 (0.8), 229.0491 (0.5), 217.0503 (0.7), 199.0393 (2.8), 151.0023 (4.7), 133.0280 (8.7), 107.0122 (2.2)	5.38	0.978	2
**73**	isorhamnetin *O*-hexuronide ^b^	C_22_H_20_O_13_	491.0832	491.0836 (72.0), 387.0720 (0.4), 357.0628 (0.7), 315.0511 (100), 300.0275 (52.0), 272.0325 (8.1), 255.0290 (0.2), 243.0295 (0.3), 229.6530 (0.2), 215.0344 (0.3) 175.0232 (1.0), 151.0025 (1.6), 107.0118 (1.0)	5.47	0.970	2
**74**	kaempferol 7-*O*-rutinoside ^b^	C_27_H_30_O_15_	593.1520	593.1519 (100), 285.0403 (74.9), 284.0325 (44.5), 255.0297 (36.6), 227.0344 (24.1), 211.0394 (1.7), 163.0022 (1.7), 151.0020 (1.5), 107.0117 (1.3)	5.65	1.124	1,2
**75**	nepetin *O*-hexoside ^b^	C_22_H_22_O_12_	477.1038	477.1041 (100), 315.0486 (33.8), 300.0268 (16.2), 299.0198 (19.1), 285.0407 (2.6), 271.0243 (2.5), 255.0304 (1.1), 243.0290 (3.2), 227.0341 (3.1), 199.0391 (8.8), 136.9868 (0.9), 133.0281 (10.1)	5.67	−0.253	1,2,3
**76**	axillarin *O*- pentosylhexoside ^b^	C_28_H_32_O_17_	639.1567	639.1567 (100), 345.0616 (73.7), 330.0387 (21.0), 315.0145 (7.7), 287.0190 (4.8)	5.74	0.012	2
**77**	apigenin *O*-pentosylhexoside ^b^	C_26_H_28_O_14_	563.1406	563.1412 (32.9), 269.0453 (100), 239.0337 (0.3), 225.0561 (1.1), 151.0022 (0.9), 117.0330 (3.6), 107.0122 (1.8)	5.75	0.961	2
**78**	apigenin 7-*O*-rutinoside^b^	C_27_H_30_O_14_	577.1570	577.1570 (48.7), 269.0454 (100), 457.1350 (1.5), 239.0348 (0.3), 225.0556 (1.5), 163.0388 (6.6), 119.0486 (10.2), 117.0330 (3.3), 107.0124 (1.9)	5.82	1.250	2
**79**	isorhamnetin 3-*O*-glucoside ^a,b^	C_22_H_22_O_12_	477.1042	477.1038 (100), 315.0493 (12.8), 314.0432 (56.2), 299.0200 (4.3), 271.0246 (23.1), 257.0453 (5.4), 243.0293 (24.6), 227.0343 (3.3), 215.0341 (3.3), 199.0391 (4.0),178.9975 (0.6), 151.0023 (3.2), 107.0122 (0.8)	5.90	0.253	1,2,3
**80**	hispidulin *O*-pentosylhexoside ^b^	C_27_H_30_O_15_	593.1512	593.1523 not fragmented *	5.93	1.832	2
**81**	isorhamnetin *O*-pentoside ^b^	C_21_H_19_O_11_	447.0935	447.0935 (100), 315.0486 (7.4), 314.0436 (43.6), 300.0276 (20.4), 285.0415 (6.3), 271.0247 (23.2), 255.0304 (2.1), 243.0294 (15.6), 227.0340 (2.6), 151.0020 (2.0)	6.02	0.437	1, 2
**82**	chrysoeriol *O*-pentosylhexoside ^b^	C_27_H_30_O_15_	593.1512	593.1530 not fragmented *	6.04	2.962	2
**83**	apigenin *O*-hexuronide ^b^	C_21_H_18_O_11_	445.0787	445.6779 (29.6), 269.0453 (100), 225.0550 (1.8), 213.0537 (0.1), 197.0596 (1.2), 183.0440 (1.3), 175.0237 (15.2), 151.0024 (2.1), 117.0330 (6.6), 107.0123 (2.9)	6.13	0.484	2
**84**	kaempferol 3-*O*-glucoside ^a,b^	C_21_H_19_O_11_	447.0935	447.0935 (100), 285.0393 (15.8), 284.0326 (51.3), 255.0296 (36.5), 227.0344 (37.4), 211.0395 (1.4), 151.0023 (2.3)	6.21	0.504	1, 2
**85**	jaceosidin *O*-hexuronide ^b^	C_23_H_22_O_13_	505.0988	505.0994 (95.1), 371.0758 (0.8), 329.0667 (100), 314.0433 (18.018), 299.0197 (35.648), 285.0405 (2.0), 271.0247 (36.4), 243.0306 (0.4), 227.0341 (1.0), 175.0236 (11.1), 161.0227 (0.9), 113.0227 (31.0), 85.0278 (19.0)	6.33	−2.731	2,3
**86**	chrysoeriol *O*-hexuronide ^b^	C_22_H_20_O_12_	475.0882	475.0883 (87.3), 299.0560 (100), 284.0325 (68.0), 256.0374 (7.3), 239.0351 (0.3), 227.0356 (1.7), 211.0387 (0.9), 175.0236 (15.1), 151.0021 (2.1), 139.0015 (0.3), 107.0125 (2.8)	6.34	0.254	2
**87**	jaceosidin *O*-hexoside	C_23_H_24_O_12_	491.1195	491.1199 (100), 329.0667 (4.8), 328.0586 (8.9), 313.0356 (35.8), 298.0136 (9.2), 285.0400 (4.3), 270.0179 (15.6), 136.9867 (1.0)	6.50	0.877	2
**88**	eupatilin *O*-hexoside	C_24_H_26_O_12_	505.1351	505.1356 not fragmented *	7.49	0.932	2
**89**	luteolin ^a^	C_15_H_10_O_6_	285.0405	285.0403 (100), 217.0495 (1.0), 199.0394 (1.8), 175.0391 (1.9), 151.0023 (4.3), 133.0280 (24.1), 121.0279 (1.1), 107.0121 (4.5)	7.58	−0.636	2,3
**90**	quercetin ^a^	C_15_H_10_O_7_	301.0354	301.0354 (100), 273.0409 (2.5), 257.0482 (0.7), 178.9975 (25.3), 151.0023 (42.4), 121.0279 (12.1), 107.0123 (12.8)	7.63	−0.019	1,2
**91**	patuletin (6-methoxyquercetin) ^b^	C_16_H_12_O_8_	331.0464	331.0460 (100), 316.0223 (64.6), 287.0198 (7.4), 271.0245 (6.2), 259.0246 (3.6), 243.0292 (2.2), 181.0134 (5.5),165.9895 (17.5), 139.0023 (11.3), 136.9863 (1.2), 121.0280 (3.1), 109.9994 (10.6)	7.72	0.149	2,3
**92**	nepetin (6-methoxyluteolin) ^b^	C_16_H_12_O_7_	315.0514	315.0514 (73.4), 300.0278 (100), 272.0317 (0.4), 255.0307 (0.6), 243.0306 (1.5), 227.0348 (1.7), 165.9895 (0.8), 139.0029 (0.7), 136.9868 (10.1), 133.0287 (2.7), 109.9997 (1.6)	7.75	1.251	1,2,3
**93**	spinacetin ^b^	C_17_H_14_O_8_	345.0616	345.0613 (100), 330.0380 (94.95), 315.0148 (30.00), 287.0196 (24.64), 259.0245 (14.99), 243.0296 (2.13), 231.0292 (2.05), 215.0341 (5.30), 187.0390 (3.78), 175.0388 (0.16), 165.9890 (0.52), 163.0387 (1.08), 149.0230 (2.98), 139.0022 (1.05), 136.9864 (1.81)	7.85	−0.726	2,3
**94**	axillarin ^b^	C_17_H_14_O_8_	345.0616	345.0615 (100), 330.0381 (99.2), 315.0147 (48.3), 287.0196 (14.5), 271.0241 (1.6), 259.0245 (3.6), 243.0294 (3.5), 231.0293 (4.8), 215.0341 (4.2), 175.0026 (3.2), 165.9894 (5.6), 149.0230 (10.0), 139.0386 (2.9), 136.9867 (1.1), 121.0281 (1.5), 109.9994 (3.4)	8.25	−0.205	2,3
**95**	apigenin ^a^	C_15_H_10_O_5_	269.0457	269.0454 (100), 225.0551 (1.0), 201.0541 (0.4), 151.0022 (5.3), 121.0124 (1.1), 117.0330 (18.4), 107.0124 (4.4)	8.62	−1.942	2
**96**	hispidulin (scutellarein-6-methyl ether) ^a,b^	C_16_H_12_O_6_	299.0563	299.0560 (65.36), 284.0323 (100), 255.0299 (1.50), 227.0340 (3.52), 211.0393 (2.15), 165.9894 (0.86), 136.9865 (15.57), 117.0329 (1.85)	8.84	−0.372	2,3
**97**	quercetagetin-3,6,3’(4’)-trimethyl ether ^b^	C_18_H_16_O_8_	359.0772	359.0776 (100), 344.0539 (85.8), 329.0305 (41.5), 314.0068 (2.8), 301.0356 (10.3), 286.0123 (7.3), 258.0169 (4.0), 242.0218 (15.0), 230.0207 (2.5), 214.0267 (9.1), 186.0303 (1.7), 163.0381 (1.7), 161.0223 (1.3), 109.9985 (0.4)	9.08	1.112	2,3
**98**	isorhamnetin^a,b^	C_16_H_12_O_7_	315.0512	315.0514 (100), 300.0278 (48.1), 271.0254 (3.3), 255.0296 (2.4), 243.0300 (1.4), 227.0340 (1.8), 211.0388 (0.5), 163.0025 (3.0), 151.0025 (8.4), 107.0124 (8.0)	9.11	−0.551	1,2
**99**	jaceosidin (6-hydroxyluteolin-6,3’-dimethyl ether) ^a,b^	C_17_H_14_O_7_	329.0677	329.0667 (87.3), 314.0433 (100), 299.0197 (20.2), 271.0249 (33.3), 255.0288 (0.6), 243.0296 (3.0), 227.0346 (2.9), 215.0347 (1.9), 199.163.0021 (1.9), 136.9868 (2.3), 135.0076 (0.6), 133.0279 (4.5)	9.15	0.073	2,3
**100**	cirsiliol^b^	C_17_H_14_O_7_	329.0677	329.0670 (100), 314.0436 (85.7), 299.0198 (35.2), 271.0250 (62.0), 243.0301 (1.0), 199.0393 (0.9), 161.0231 (0.8), 151.0028 (0.5)	9.47	0.954	2,3
**101**	quercetagetin-3,6,3’(4’)-trimethyl ether ^b^	C_18_H_16_O_8_	359.0772	359.0777 (100), 344.0539 (42.5), 329.0306 (63.0), 314.0073 (14.7), 301.0346 (1.6), 286.0122 (3.5), 258.0172 (4.3), 230.0216 (1.5), 214.0269 (0.3), 202.0258 (1.2), 165.9889 (0.4), 163.0391 (7.6), 148.0153 (10.2), 139.0019 (0.5), 136.9864 (1.2)	9.66	1.196	2,3
**102**	cirsimaritin (6-hydroxyapigenin-6,7-dimethyl ether) ^a,b^	C_17_H_14_O_6_	313.0719	313.0718 (100), 298.0480 (38.6), 283.0248 (12.6), 269.0455 (12.8), 255.0295 (35.2), 227.0337 (0.4), 151.0017 (0.4), 107.0122 (0.5)	10.39	0.059	2,3
**103**	eupatilin/santin ^b^	C_18_H_16_O_7_	343.0812	343.0824 (100), 328.0591 (69.9), 313.0358 (47.5), 298.0117 (15.9), 285.0411 (2.4), 270.0173 (11.6), 242.0221 (4.8), 214.0266 (2.1), 163.0029 (4.4), 147.0438 (4.7), 136.9864 (2.3), 132.0203 (4.1), 109.9997 (0.4)	10.68	−0.047	2,3

^a^ Compared to a reference standard; ^b^ reported for the first time; 1-*T. balsamita* flower heads; 2-*T. balsamita* leaves; 3-*T. balsamita* roots; * annotation was done in (+) ESI-MS/MS (see Appendix A).

**Table 3 plants-12-00022-t003:** Antioxidant activity of the *T. balsamita* extracts.

Samples	PMD Assay (mmol TE/g)	DPPH (mg TE/g)	ABTS (mg TE/g)	CUPRAC (mg TE/g)	FRAP (mg TE/g)	Metal Chelating (mg EDTAE/g)
Leaves	1.09 ± 0.03 ^c^	43.87 ± 0.26 ^b^	65.64 ± 1.77 ^c^	86.71 ± 0.72 ^c^	57.34 ± 0.08 ^b^	36.16 ± 0.36 ^a^
Roots	1.48 ± 0.01 ^a^	44.87 ± 0.08 ^b^	91.52 ± 0.76 ^b^	137.08 ± 0.55 ^b^	92.21 ± 2.05 ^a^	33.00 ± 1.18 ^b^
Flower heads	1.20 ± 0.03 ^b^	84.54 ± 3.35 ^a^	96.35 ± 2.22 ^a^	151.20 ± 0.22 ^a^	93.22 ± 1.59 ^a^	17.43 ± 1.87 ^c^

Values are reported as mean ± SD of three parallel experiments. TE: Trolox equivalent; EDTAE: EDTA equivalent. Different letters indicate significant differences in the tested extracts (*p* < 0.05).

**Table 4 plants-12-00022-t004:** Enzyme inhibitory activity of the *T. balsamita* extracts.

Samples	AChE (mg GALAE/g)	BChE (mg GALAE/g)	Tyrosinase (mg KAE/g)	α-Amylase (mmol ACAE/g)	α-Glucosidase (mmol ACAE/g)	Lipase (mg OE/g)
Leaves	2.11 ± 0.04 ^a^	2.43 ± 0.04 ^a^	54.65 ± 1.30 ^a^	0.44 ± 0.01 ^a^	0.19 ± 0.05 ^c^	4.02 ± 0.67 ^b^
Roots	2.00 ± 0.03 ^b^	1.33 ± 0.20 ^b^	51.43 ± 0.66 ^b^	0.43 ± 0.02 ^a^	0.71 ± 0.07 ^a^	8.15 ± 1.00 ^a^
Flower heads	1.83 ± 0.08 ^c^	na	45.49 ± 1.11 ^c^	0.28 ± 0.02 ^b^	0.50 ± 0.03 ^b^	na

Values are reported as mean ± SD of three parallel experiments. GALAE: Galantamine equivalent; KAE: Kojic acid equivalent; ACAE: Acarbose equivalent; OE: Orlistat equivalent; na: not active. Different letters indicate significant differences in the tested extracts (*p* < 0.05).

## Data Availability

Not applicable.

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
