# Peer review of "An In-Depth Study of Metabolite Profile and Biological Potential of Tanacetum balsamita L. (Costmary)"

_plants, 2022, doi:10.3390/plants12010022_

Round 1

Reviewer 1 Report

This article revealed metabolite profile and biological potential of Tanacetum balsamita L. (costmary). The analysis is useful to understand metabolite composition and biological applications of medicinal plants. Before recommending this article for publication, there are some shortcomings for that should be resolve.

Abstract is well written however quantitative results should be provide for biological activities.

Add conclusion and future recommendation of the study in the abstract.

In introduction first paragraph should be start from general discussion about medicinal or traditional plants.

Provide origin of the T. balsamita

Line 69, No need of the “As Asteraceae taxa”

Line 530, sentence should not be start with numbers.

Add subsection in section 3.

Provide complete protocols of every method.

Determination of antioxidant and enzyme inhibitory effects” could be cited with

https://doi.org/10.1007/s10534-022-00417-1,

Cytotoxic assay should be cited with relevant study

http://doi.org/10.36899/JAPS.2022.3.0484,

Conclusion is well presented however practical applications of the species based on these results should be provided.

Author Response

We would like to thank for the valuable comments, which gave us a chance to improve quality of our paper and correct some errors. We do hope that modifications included in the new version of manuscript will meet your approval. All changes and corrections have been clearly marked in the revised text. 

To facilitate the comprehension of this letter, the comments were copied in red and the responses were written immediately after the comments.

Abstract is well written however quantitative results should be provide for biological activities.

The abstract was revised; quantitative results on the biological activity were embedded.

The flower heads extract possessing the highest content of total phenolics and flavonoids, actively scavenged DPPH (84.54 ± 3.35 mgTE/g) and ABTS radicals (96.35 ± 2.22 mgTE/g), and showed the highest reducing potential (151.20 and 93.22 mg TE/g for CUPRAC and FRAP, respectively). The leaves extract exhibited the highest inhibition towards acetyl- and butyrylcholinesterase (2.11 and 2.43 mg GALAE/g, respectively) and tyrosinase (54.65 mg KAE/g). The root extract inhibited α-glucosidase (0.71 ± 0.07 mmol ACAE/g), α-amylase (0.43 ± 0.02 mmol ACAE/g) and lipase (8.15 ± 1.00 mg OE/g). At concentration > 2µg/mL, a significant dose dependent reduction of cell viability towards THP-1 monocyte leukemic cells was observed. Costmary could be recommended for raw material production with antioxidant and enzyme inhibitory properties.

Add conclusion and future recommendation of the study in the abstract.

According to the reviewer’s suggestion, a new sentence was embedded.

Costmary could be recommended for raw material production with antioxidant and enzyme inhibitory properties.

In introduction first paragraph should be start from general discussion about medicinal or traditional plants.

A new sentence was embedded into the text:

The use of plants as sources of drugs and secondary metabolites have been attracting scientific attention over the past decades, considering not only the well known medicinal species but also plants used in traditional medicines and a variety of edible plants.

Provide origin of the T. balsamita

The plant origin was provided in the subsection 3.1. as follows:

The seedlings were provided by the greenhouse “Zelena prolet” (Sofia, Bulgaria).

Line 69, No need of the “As Asteraceae taxa”

The change was done.

Line 530, sentence should not be start with numbers.

The change was make.

Add subsection in section 3.

The change was done.

Provide complete protocols of every method.

Appropriate references have been added to support the protocol used. For the sake of similarity, we have not included all details of the protocols as these have been previously published. However, all experimental details have been given in the supplemental materials

Determination of antioxidant and enzyme inhibitory effects” could be cited with

https://doi.org/10.1007/s10534-022-00417-1,

The reference has been inserted in the revised version.

Cytotoxic assay should be cited with relevant study.

http://doi.org/10.36899/JAPS.2022.3.0484,

The reference has been inserted in the revised version.

Conclusion is well presented however practical applications of the species based on these results should be provided.

According to the reviewer’s suggestions, new sentences were embedded.

In addition to evoking an antioxidant response, costmary extracts display in vitro enzyme inhibitory effects, which generates interest in the plant as a valuable herbal drug. Moreover, this study advocates further work geared towards additional in vivo studies.

Reviewer 2 Report

This study is a comprehensive characterization of an aromatic and medicinal plant Tanacetum balsamita. This herb is quite popularfor its pleasantly scented essential oil, so most of previously published literature concerns the volatile compounds. On ly a few previous reports exist to show a significant content of polyphenols. However, the present contribution is the first one to use high resolution tandem MS to detect and annotate the full profile of extracts from three plant parts – inflorescences (flowerheads), leaves and roots. In addition, several in vitro chemical, enzymatic and cell line bioassays were performed to indicate potential in further bioactivity research.

The applied methods are appropriate for such subject and were performed properly. The LC-MS orbitrap part is a dominant one and is painstakingly described and interpreted in detail. It is the strongest part of the submission. A lot of chromatographic and spectral data obtained are provided either in the main text or in the supplementary file. It is of great value for other researchers. On the other hand, the bioactivity screening used routine and quite preliminary tests but it can be considered as just indicative studies.

Therefore, the paper is worth publishing. Unfortunately, the present submission has several shortcomings which have to be improved before acceptance.

1. The most striking is the uneven level of English, some parts are correct but some are full of mistakes (like there were several authors writing independently). One example is an odd overuse of "witnessed" throughout. So, please, put it in order, preferably by rewriting the manuscript in a coherent style. Thereafter, secure a thorough editing by a proficient English writer or, better yet, a qualified native speaker.

2. The claim that inhibition of digestive enzymes (α-glucosidase, α-amylase, lipase) is a proff of beneficial action against metabolic disorders is too far going and should be avoided, even in the conclusions.

3. Also, when describing the content or activity use flowerhead/leaf/root EXTRACTS, not leaves, roots, flowerheads.

4. Line 153 – correct "acid acid"; Line 167 – should read "hydroxycinnamic acids"; line 277 – "profiles" not "profiling"; line 290 – quercetagenin "derivatives" not "descendants";

5. Rephrase sentence in lines 345-347; line 350 – "leaflets"?; line 352 – "promoted"?

6. Line 353-354 – the whole sentence is nonsense!

7. In figure 1 – use flowerheads instead of flowers. The image is low resolution and the lower row legend is totally unreadable – make it bigger please;

8. The information in lines 374-380 is confusing. PLease rewrite it in a more straightforward way;

9. Lines389-407 – these values (bioactivity) should be discussed in a broader context, not only compared to related species – are these values pharmacologically relevant (honestly, don't seem so)?

10. Figure 2 is hardly readable. Please, make it larger, high-res or better yet, consider rearranging it. Also, the part C needs completing (only dots are there);

11. Reinterpret section 2.7. It is not clear to read and the value for roots at 1000 μg/mL is odd. Line 462-463 – this sentence is redundant and can be deleted; Use flowerheads, not flowers in the Fig. 3 legend. The Y axis is not cytotoxicity! Why control values are separate and different for roots/flheads and leaves, when there were no extracts added?

12. What was the fresh mass/dried material ratio?

13. In the conclusions, try to go beyond the trivial statement that these results can lead to use in the industries based on the in vitro enzyme-inhibition (being in some cases not very strong). For example, the chemotaxonomic significance of the results and some differences from the previous studies (absence of cichoric acid) can be considered.-

Author Response

We would like to thank for the valuable comments, which gave us a chance to improve quality of our paper and correct some errors. We do hope that modifications included in the new version of manuscript will meet your approval. All changes and corrections have been clearly marked in the revised text.

To facilitate the comprehension of this letter, the comments were copied in red and the responses were written immediately after the comments.

  1. The most striking is the uneven level of English, some parts are correct but some are full of mistakes (like there were several authors writing independently). One example is an odd overuse of "witnessed" throughout. So, please, put it in order, preferably by rewriting the manuscript in a coherent style. Thereafter, secure a thorough editing by a proficient English writer or, better yet, a qualified native speaker.

The changes were made throughout the text. The English was carefully checked and revised.

  1. The claim that inhibition of digestive enzymes (α-glucosidase, α-amylase, lipase) is a proff of beneficial action against metabolic disorders is too far going and should be avoided, even in the conclusions.

According to the reviewer’s suggestion, the abstract was rewritten and becomes now:

The flower heads extract possessing the highest content of total phenolics and flavonoids, actively scavenged DPPH (84.54 ± 3.35 mgTE/g) and ABTS radicals (96.35 ± 2.22 mgTE/g), and showed the highest reducing potential (151.20 and 93.22 mg TE/g for CUPRAC and FRAP, respectively). The leaves extract exhibited the highest inhibition towards acetyl- and butyrylcholinesterase (2.11 and 2.43 mg GALAE/g, respectively) and tyrosinase (54.65 mg KAE/g). The root extract inhibited α-glucosidase (0.71 ± 0.07 mmol ACAE/g), α-amylase (0.43 ± 0.02 mmol ACAE/g) and lipase (8.15 ± 1.00 mg OE/g). At concentration > 2µg/mL, a significant dose dependent reduction of cell viability towards THP-1 monocyte leukemic cells was observed. Costmary could be recommended for raw material production with antioxidant and enzyme inhibitory properties.

  1. Also, when describing the content or activity use flowerhead/leaf/root EXTRACTS, not leaves, roots, flower heads.

The changes were made throughout the text.

  1. Line 153 – correct "acid acid"; Line 167 – should read "hydroxycinnamic acids"; line 277 – "profiles" not "profiling"; line 290 – quercetagenin "derivatives" not "descendants";

The changes were made throughout the text.

  1. Rephrase sentence in lines 345-347; line 350 – "leaflets"?; line 352 – "promoted"?

The sentence was rephrased:

“Similarly, higher concentration of the group A2 metabolites in the flower heads extract was found, while lower levels of the group E metabolites were established”.

The changes were made.

  1. Line 353-354 – the whole sentence is nonsense!

The sentence was deleted.

  1. In figure 1 – use flowerheads instead of flowers. The image is low resolution and the lower row legend is totally unreadable – make it bigger please;

The picture resolution was improved.

  1. The information in lines 374-380 is confusing. PLease rewrite it in a more straightforward way;

The paragraph was revised and becomes now:

Generally, the received data for radical scavenging activity are consistent with those recorded previously in T. parthenium, T. poteriifolium and T. vulgare extracts and substantially lower in comparison with T. macrophyllum extracts [20, 21, 25, 26]. The aforementioned Tanacetum species revealed higher reducing power activity than costmary extracts. T. balsamita leaves extract possessed strong chelating ability being more potent compared to T. parthenium, T. poteriifolium and T. vulgare extracts.

  1. Lines389-407 – these values (bioactivity) should be discussed in a broader context, not only compared to related species – are these values pharmacologically relevant (honestly, don't seem so)?

            According to the reviewer’s suggestions, the following paragraph was embedded:

            For instance, flavonoids and acylquinic acids have been shown as inhibitors of the studied enzymes. However, the enzyme inhibitory potential is not directly related to TPC and TFC as has been seen in T. vulgare and T. macrophyllum [20, 21].

            It appears that the enzyme inhibition could be ascribed to the non—phenolic compounds in the assayed extracts such as sesquiterpene lactones. With this respect, sesquiterpene lactones act probably in a synergistic way in AChE related disorders [42]. Indeed, the germacranolide parthenolide and monoterpene thujone have been already reported as cholinesterase inhibitors [43-45]. Orhan et al. (2015) hypothesized that parthenolode plays a role in AChE inhibition in a synergistic manner together with other compounds (monoterpenes). Thus, the leaf extract of Tanacetum argenteum subsp. flabellifolium had the highest AChE inhibitory effect (96.68 ± 0.35%). C-flavonoid glycoside homoorientin, identified in cosmary leaves extract, was previously reported to inhibit AChE in an in silico and in vivo study (Rasool et al. 2018). At 100 mg/kg for 3 weeks homoorientin inhibited the activity of AChE in rats with experimentally induced Alzheimer’s disease. Hispidulin, identified in the Phyla nodiflora extracts, was previously reported to exhibit tyrosinase with an IC50 value of 146 µM (Lin et al, 2014). In addition, chlorogenic acid and its derivatives have potential as cholinesterase and glucosidase inhibitors [46-48]. Thus, chlorogenic acid inhibited AChE and BChE and pro-oxidant-induced lipid peroxidation in rat brain in vitro (IC50 value of 8.01 mg/ml and 6.3 mg/ml, respectively) (47). At 5 mg/kg 3,5-dicaffeoylquinic acid reduced significantly the blood glucose levels and ameliorate the oxidative stress biomarkers reduced glutathione, malondialdehyde, and serum biochemical parameters (48).

  1. Figure 2 is hardly readable. Please, make it larger, high-res or better yet, consider rearranging it. Also, the part C needs completing (only dots are there);

The changes were made.

  1. Reinterpret section 2.7. It is not clear to read and the value for roots at 1000 μg/mL is odd. Line 462-463 – this sentence is redundant and can be deleted; Use flower heads, not flowers in the Fig. 3 legend. The Y axis is not cytotoxicity! Why control values are separate and different for roots/fl. heads and leaves, when there were no extracts added?

The text was rephrased “We reach a toxicity of 50% at 200 µg mL-1 for flower heads extract, 1000 µg mL-1 and 2000 µg mL-1 for leaves and roots extracts, respectively, which is to be considered as a very high concentration, without any biological meaning. "

In fact, value for roots at 1000 μg/mL could be considered as artefactual.

Line 462-3 - The sentence was omitted.

Y axis legend is Viability (%).

To remove the interactions between the WST-1 test and the plant extracts, each concentration was worked out by subtracting the mix medium+plant extract.

Second, to permit a better comparison between each extract, the 100% of viability was worked out based on the viability measured for roots extract, explaining the slight shift at 0 µg/mL concentration.

  1. What was the fresh mass/dried material ratio?

If you mean extraction procedure, (1:20 w/v), 1 g dry plant material was extracted with 20 ml 80% methanol.

  1. In the conclusions, try to go beyond the trivial statement that these results can lead to use in the industries based on the in vitro enzyme-inhibition (being in some cases not very strong). For example, the chemotaxonomic significance of the results and some differences from the previous studies (absence of cichoric acid) can be considered.-

In response to the reviewer’s suggestions, the following paragraph was embedded into the Conclusion section:

Despite of previously published data on the high concentration of cichoric acid in the cosmary aerial parts extract, we were not able to confirm the presence of either cichoric acid or and any esters of tartaric acid and hydroxycinnamic acids. According to this study, the presence of 6-methoxylated flavones and flavonols, dicaffeoylquinic acids and their hexosides and phenolic acids glycosides could be considered significant in the chemotaxonomy of the Tanacetum genus.

Reviewer 3 Report

Line 20: “in vitro” italic!

Line 76: Rewrite “Diosmetin and acacetin glycosides were reported as well as [5].”

Table 1: Use the abbreviations introduced in line 103 (TPC and TFC) rather than the full words. Include the unit for the total flavonoid content in column 3: mg QE/g. Moreover, use TAC rather than PMD assay for the last column.

Line 99: Results and Discussions ??

Lines 128-129: Merge this paragraph with the next one. They are connected!

Line 130 and 140: You explain this to be a non-targeted analysis, but then talk about targeted analysis. This is confusing. The aim is to analyse hydroxycinnamic acids and flavonoids. Then we have moved from a non-targeted analysis into a narrower zone. But you can choose to define this as a non-targeted analysis of hydroxycinnamic acids and flavonoids. But again the term ‘flavonoids’ is quite wide (>10.000 structures..), but you analyse flavonols and flavones. Please clarify and make a choice.

Paragraph 2.1.1 does not contain any results, and thus belongs under the heading of Methods.

Line 153: Remove on ‘acid’

Line 154: Erase ‘assayed’

Line 168: Change ‘evidenced’ to e.g. ‘analysed’ or ‘revealed’.

Line 216: When talking about ‘peaks’ we would like to see them. If no figure of the chromatogram is present rather use the word ‘compound’.

Line 240: [Nar-H]- is not a correct term/format

Line 246: ‘confirmed’ rather than ‘evidenced’

Line 296: ‘additional’

Line 299: remove ‘a’

Line 301: exchange ‘witnessed’

Table 2 compound 88: Remove the number 7 (?)

Paragraph 2.3 This is based on qualitative analysis, and not quantitative data. This makes the whole theme less interesting. The lower print of the heat map is unreadable. I would recommend to leave this paragraph out.

Table 3: How is this connected with the last column found in table 1? You use two decimal digits. Are these digits significant? I doubt so!

Table 4: Check numbers of significance.

Figure 2: Hard to read due to small font sizes. Improve!

Line 507: Change to ‘..as mobile phases.’

Line 511: Change reference to a reference number [21]?

Line 554: ‘Chlorogenic, 3, 5-diCQA and 4, 5-diCQA acid were the main compounds in the leaves and roots.’ and line 556 claiming ‘..rich in rutin’. How can you say without quantification?

References: Journal abbreviations have to be used. In some cases this have been done, in others not.

Author Response

We would like to thank for the valuable comments, which gave us a chance to improve quality of our paper and correct some errors. We do hope that modifications included in the new version of manuscript will meet your approval. All changes and corrections have been clearly marked in the revised text.

To facilitate the comprehension of this letter, the comments were copied in red and the responses were written immediately after the comments.

Line 20: “in vitro” italic!

The change was done.

Line 76: Rewrite “Diosmetin and acacetin glycosides were reported as well as [5].”

The change was done.

Table 1: Use the abbreviations introduced in line 103 (TPC and TFC) rather than the full words. Include the unit for the total flavonoid content in column 3: mg QE/g. Moreover, use TAC rather than PMD assay for the last column.

The changes were done. The table was modified and the column 4 was embedded into the table 3 (Antioxidant activity).

Line 99: Results and Discussions ??

The change was done.

Lines 128-129: Merge this paragraph with the next one. They are connected!

The change was done.

Line 130 and 140: You explain this to be a non-targeted analysis, but then talk about targeted analysis. This is confusing. The aim is to analyse hydroxycinnamic acids and flavonoids. Then we have moved from a non-targeted analysis into a narrower zone. But you can choose to define this as a non-targeted analysis of hydroxycinnamic acids and flavonoids. But again the term ‘flavonoids’ is quite wide (>10.000 structures..), but you analyse flavonols and flavones. Please clarify and make a choice.

According to the reviewer’s suggestion, the sentence was revised as follows:

To assess the secondary metabolites, non-targeted metabolic profiling of the hydroxycinnamic acids, flavones, flavonols and flavanones of each methanol-aqueous extract was carried out by UHPLC-Orbitrap-HRMS.

Paragraph 2.1.1 does not contain any results, and thus belongs under the heading of Methods.

The paragraph was moved in the Methods (subsection 3.5.)

Line 153: Remove on ‘acid’

The change was done.

Line 154: Erase ‘assayed’

The change was done.

Line 168: Change ‘evidenced’ to e.g. ‘analysed’ or ‘revealed’.

The change was done.

Line 216: When talking about ‘peaks’ we would like to see them. If no figure of the chromatogram is present rather use the word ‘compound’.

The change was done.

Line 240: [Nar-H]- is not a correct term/format

The sentence becomes now “Additionally, the aglycone naringenin in 61 was evidenced by the deprotonated molecule at m/z 271.062 supported by the RDA ions at m/z 163.003 (0,2A-), 151.002 (1,3A-), 119.049 (1,3B-) and 107.012. (0,4A-).

Line 246: ‘confirmed’ rather than ‘evidenced’

The change was done.

Line 296: ‘additional’

The change was done.

Line 299: remove ‘a’

The change was done.

Line 301: exchange ‘witnessed’

The change was done.

Table 2 compound 88: Remove the number 7 (?)

The change was done.

Paragraph 2.3 This is based on qualitative analysis, and not quantitative data. This makes the whole theme less interesting. The lower print of the heat map is unreadable. I would recommend to leave this paragraph out.

Response: Thank you for your question. We performed the analysis using peak areas for each compound, thus determining the differences between the plant parts tested.  From this figure, clearly the distribution of metabolites changed by tested plant parts. We have tried to improve the quality of the heat map and it is now maximum resolution.

Table 3: How is this connected with the last column found in table 1? You use two decimal digits. Are these digits significant? I doubt so!

Response: We have revised the parts and we have moved the total antioxidant assays results in Table 3.

Table 4: Check numbers of significance.

Response: We have checked and added the significance.

Figure 2: Hard to read due to small font sizes. Improve!

Response: We have improved the figure quality

Line 507: Change to ‘..as mobile phases.’

The change was done.

Line 511: Change reference to a reference number [21]?

The change was done.

Line 554: ‘Chlorogenic, 3, 5-diCQA and 4, 5-diCQA acid were the main compounds in the leaves and roots.’ and line 556 claiming ‘..rich in rutin’. How can you say without quantification?

The text was revised and becomes now “Chlorogenic, 3, 5-diCQA and 4, 5-diCQA acid dominated the leaves and roots profiles. To understand the relationship between plant parts and biological activity, multivariate statistical analyses were performed. The strongest antioxidant activity (DDPH, FRAP, CUPRAC and ABTS) of the flower heads extract could be related to the presence of rutin, isoquercitrin and hyperoside, and the corresponding aglycone.”

References: Journal abbreviations have to be used. In some cases this have been done, in others not.

All references have been checked and corrected again.